# Underestimation of Anthropogenic Organosulfates in Atmospheric Aerosols in Urban Regions

Yanting Qiu[1#], Junrui Wang[1#], Tao Qiu[2], Jiajie Li[3], Yanxin Bai[4], Teng Liu[1], Ruiqi Man[1], Taomou Zong[1], Wenxu Fang[1], Jiawei Yang[1], Yu Xie[1], Zeyu Feng[1], Chenhui Li[3], Ying Wei[3], Kai Bi[5], Dapeng Liang[2], Ziqi Gao[6], Zhijun Wu[1,7,*], Yuchen Wang[4,*], Min Hu[1]

[1] State Key Laboratory of Regional Environment and Sustainability, International Joint Research Center for Atmospheric Research (IJRC), College of Environmental Sciences and Engineering, Peking University, Beijing, China

[2] Key Lab of Groundwater Resources and Environment of the Ministry of Education, College of New Energy and Environment, Jilin University, Changchun, China

[3] College of Environment and Ecology, Laboratory of Compound Air Pollutions Identification and Control, Taiyuan University of Technology, Taiyuan, China

[4] College of Environmental Science and Engineering, Hunan University, Changsha, China

[5] Beijing Weather Modification Center, Beijing Meteorological Service, Beijing, China

[6] University of Virginia Environmental Institute, Charlottesville, United States

[7] Collaborative Innovation Center of Atmospheric Environment and Equipment Technology, Nanjing University of Information Science and Technology, Nanjing, China

#: Yanting Qiu and Junrui Wang contributed equally to this work

*Correspondence Author:

Zhijun Wu (zhijunwu@pku.edu.cn) and Yuchen Wang (ywang@hnu.edu.cn)

## ABSTRACT

Organosulfates (OSs) are important components of organic aerosols, which serve as critical tracers of secondary organic aerosols (SOA). However, molecular composition, the relationship between OSs and their precursors, and formation driving factors of OSs at different atmospheric conditions have not been fully constrained. In this work, we integrated OS molecular composition, precursor-constrained positive matrix factorization (PMF) source apportionment, and OS-precursor correlation analysis to classify OS detected from $PM_{2.5}$ samples according to their volatile organic compounds (VOCs) precursors collected from three different cities (Beijing, Taiyuan, and Changsha) in China. This new approach enables the accurate classification of OSs from molecular perspective. Compared with conventional classification methods, we found the mass fraction of Aliphatic OSs (including nitrooxy OSs; NOSs) increased by 22.0%, 17.8%, and 10.3% in Beijing, Taiyuan, and Changsha, respectively, highlighting the underestimation of Aliphatic OSs in urban regions. The formation driving factors of Aliphatic OSs during the field campaign were further investigated. We found that elevated aerosol liquid water content promoted the formation of Aliphatic OSs only when aerosols transition from non-liquid state to liquid state. In addition, enhanced inorganic sulfate mass concentrations, and $O_x$ ($O_x = NO_2 + O_3$) concentrations, as well as decreased aerosol pH commonly facilitated the formation of Aliphatic OSs. These results reveal a significant underestimation of OSs derived from anthropogenic emissions in wintertime, particularly Aliphatic OSs, highlighting the need for a deeper understanding of SOA formation and composition in urban environments.

**KEY WORDS**: organosulfate; non-target analysis; high-resolution mass spectrometry; secondary organic aerosol; PMF source apportionment

## 1. Introduction

Due to the diversity of natural and anthropogenic emissions and the complexity of atmospheric chemistry, investigating the chemical characterization and formation mechanisms of secondary organic aerosols (SOA) remains challenging. Among SOA components, organosulfates (OSs) have emerged as key tracers (Brüggemann et al., 2020; Hoyle et al., 2011), as their formation is primarily governed by secondary atmospheric processes. Moreover, OS significantly influence the aerosol physicochemical properties, including acidity (Riva et al., 2019; Zhang et al., 2019), hygroscopicity (Estillore et al., 2016; Ohno et al., 2022; Hansen et al., 2015), and light-absorption properties (Fleming et al., 2019; Jiang et al., 2025). Therefore, a deeper understanding of OS abundance, sources, and formation drivers is crucial for elucidating SOA formation and its properties.

Quantifying OS abundance is critical to assess their contribution to SOA. However, this is difficult due to the large number and structural diversity of OSs molecules and the lack of authentic standards. Most studies quantify a few representative OSs using synthetic or surrogate standards (Wang et al., 2020; Wang et al., 2017; Huang et al., 2018b; He et al., 2022), while non-target analysis (NTA) with high-resolution mass spectrometry (HRMS) offers broader molecular characterization (Huang et al., 2023a; Wang et al., 2022b; Cai et al., 2020). Although NTA combined with surrogate standards allows molecular-level (semi-)quantification, overall OS mass concentration remain underestimated, and many OSs remain unidentified (Lukács et al., 2009; Cao et al., 2017; Tolocka and Turpin, 2012; Ma et al., 2025).

Classifying OS based on their precursors is a powerful approach for understanding OS formation from a mechanistic perspective. OSs from specific precursors generally share similar elemental compositions, with characteristic ranges of C atoms, double bond equivalents (DBE), and aromaticity equivalents (Xc). For example, isoprene-derived OSs typically contain 4–5 C atoms; monoterpene- and sesquiterpene-derived OSs usually have 9–10 and 14–15 C atoms, respectively (Lin et al., 2012; Riva et al., 2016c; Wang et al., 2019a; Surratt et al., 2008; Riva et al., 2015). An "OS precursor map," correlating molecular weight and carbon number based on chamber studies, has been developed to classify OSs accordingly (Wang et al., 2019a). However, these approaches often oversimplify OS formation by relying solely on elemental composition, leaving many OSs without identified precursors.

The formation mechanisms of OS remain incompletely understood, though several driving factors have been identified through controlled chamber experiments and ambient observations. For instance, increased aerosol liquid water content (ALWC) enhances OS formation by promoting the uptake of gaseous precursors (Xu et al., 2021a; Wang et al., 2021b). Inorganic sulfate can also affect OS formation by acting as nucleophiles via epoxide pathway (Eddingsaas et al., 2010; Wang et al., 2020). However, meteorological conditions vary across cities, meaning the relative importance of these factors may differ by location. Thus, evaluating these formation drivers under diverse atmospheric conditions is essential. Identifying both common and region-specific drivers is key to a comprehensive understanding of OS formation mechanisms.

In this study, we employed NTA using ultra-high performance liquid chromatography (UHPLC) coupled with high-resolution mass spectrometry (HRMS) to characterize OS molecular composition in $PM_{2.5}$ samples from three cities. Identified OSs were classified by their VOCs precursors, including

aromatic, aliphatic, monoterpene, and sesquiterpene VOCs, via precursor-constrained positive matrix
factorization (PMF). Mass concentrations were quantified or semi-quantified using authentic or
surrogate standards. Additionally, spatial variations in OS mass concentrations and environmental
conditions were analyzed to distinguish both common and site-specific drivers of OS formation.

## 2. Methodology

### 2.1 Sampling and Filter Extraction

Field observations were conducted during winter (December 2023 to January 2024) at three urban
sites in China: Beijing, Taiyuan, and Changsha. The site selection was based on contrasts in winter
meteorological conditions and dominate $PM_{2.5}$ sources. For meteorological conditions, Beijing and
Taiyuan represent northern Chinese cities with cold, dry conditions (low RH). In comparison,
Changsha is characterized by relatively higher winter RH. In terms of $PM_{2.5}$ sources, Taiyuan is a
traditional industrial and coal-mining base, Changsha's pollution profile is more influenced by traffic
and domestic cooking emissions, whereas Beijing is characterized by a high mass fraction of
secondary aerosols. This enables a comparative analysis of OS formation mechanisms under varied
atmospheric conditions. In Beijing, $PM_{2.5}$ samples were collected at the Peking University Atmosphere
Environment Monitoring Station (PKUERS; 40.00°N, 116.32°E), as detailed in previous studies
(Wang et al., 2023a). Sampling in Taiyuan and Changsha took place on rooftops at the Taoyuan
National Control Station for Ambient Air Quality (37.88°N, 112.55°E) and the Hunan Hybrid Rice
Research Center (28.20°N, 113.09°E), respectively (see Figure S1).
Daily $PM_{2.5}$ samples were collected on quartz fiber filters ($\varphi$ = 47 mm, Whatman Inc.) from 9:00
to 8:00 local time the next day. All quartz fiber filters were pre-baked at 550 ℃ for 9 hours before
sampling to remove the background organic matters. In Beijing and Taiyuan, RH-resolved sampling
was performed using a home-made RH-resolved sampler, stratifying daily samples into low (RH ≤
40%), moderate (40% < RH ≤ 60%), and high (RH > 60%) RH regimes with the sampling flow rate
of 38 L/min. Due to persistently high RH in Changsha, a four-channel sampler (TH-16, Wuhan
Tianhong Inc.) collected $PM_{2.5}$ samples without RH stratification with the flow rate of 16.7 L/min.
Consequently, Beijing and Taiyuan collected one or more samples daily, whereas Changsha collected
one sample per day. A total of 40, 64, and 30 samples were obtained from Beijing, Taiyuan, and
Changsha, respectively. The samples were stored in a freezer at -18 ℃ immediately after collection.
The maximum duration between the completion of sampling and the start of chemical analysis was
approximately 40 days. Prior to analysis, all samples were equilibrated for 24 hours under controlled
temperature (20 ± 1 ℃) and RH (40-45%) within a clean bench, in order to allow the filters to reach
a stable, reproducible condition for subsequent handling and to minimize moisture condensation.
Average daily $PM_{2.5}$ mass concentrations and RH during sampling are summarized in Table S1.
Sample extraction followed established protocols (Wang et al., 2020). Briefly, filters were
ultrasonically extracted twice for 20 minutes. A total volume of 10 mL of LC-MS grade methanol
(Merck Inc.) was used for each sample. All extracts were filtered through 0.22 μm PTFE syringe filters,
and evaporated under a gentle stream of high-purity $N_2$ (>99.99%). The dried extracts were then
redissolved in 2 mL of LC-MS grade methanol for analysis. This step was necessary to achieve
sufficient sensitivity for the detection of OSs with low concentration.
During the campaign, gaseous pollutants ($SO_2$, $NO_2$, $O_3$, CO) were monitored using automatic
analyzers. $PM_{2.5}$ and $PM_{10}$ mass concentrations were measured by tapered element oscillating
microbalance (TEOM). Water-soluble ions ($Na^+$, $NH_4^+$, $K^+$, $Mg^{2+}$, $Ca^{2+}$, $Cl^-$, $NO_3^-$, $SO_4^{2-}$) were
analyzed with the Monitor for AeRosols and Gases in ambient Air (MARGA) coupled with ion
chromatography. Organic carbon (OC) and elemental carbon (EC) were quantified by online OC/EC
analyzers or carbon aerosol speciation systems. Trace elements in $PM_{2.5}$ were determined by X-ray
fluorescence spectrometry (XRF). Additionally, VOCs concentrations were measured using an online
gas chromatography-mass spectrometry (GC-MS) system with a one-hour time resolution in Taiyuan
and Changsha. Table S2 summarizes the monitoring instruments deployed at each site. All instruments
were calibrated to ensure the reliability of the measurement data. Specifically, the online gas pollutants
and particulate matter automatic analyzers underwent automatic zero/span checks every 24 hours at
0:00 local time. For MARGA-ion chromatography, OC/EC analyzers, and XRF systems were
calibrated weekly. The online GC-MS system was automatically calibrated every 24 hours using
standard VOCs mixture.

## 2.2 Identification of Organosulfates

The molecular composition of $PM_{2.5}$ extracts was analyzed using an ultra-high performance
liquid chromatography (UHPLC) system (Thermo Ultimate 3000, Thermo Scientific) coupled with an
Orbitrap HRMS (Orbitrap Fusion, Thermo Scientific) equipped with an electrospray ionization (ESI)
source operating in negative mode. Chromatographic separation was achieved on a reversed-phase
Accucore C18 column (150 × 2.1 mm, 2.6 μm particle size, Thermo Scientific). For tandem MS
acquisition, full MS scans ($m/z$ 70–700) were collected at a resolving power of 120,000, followed by
data-dependent MS/MS ($ddMS^2$) scans ($m/z$ 50–500) at 30,000 resolving power. Detailed UHPLC-
$HRMS^2$ parameters are provided in Text S1.
NTA was performed using Compound Discoverer (CD) software (version 3.3, Thermo Scientific)
to identify chromatographic peak features (workflow details in Table S3). Molecular formulas were
assigned based on elemental combinations $C_cH_hO_oN_nS_s$ (c = 1–90, h = 1–200, o = 0–20, n = 0–1, s =
0–1) within a mass tolerance of 0.005 Da with up to one $^{13}C$ isotope. Formulas with hydrogen-to-
carbon (H/C) ratios outside 0.3–3.0 and oxygen-to-carbon (O/C) ratios beyond 0–3.0 were excluded
to remove implausible assignments. We calculated the double bond equivalent (DBE) and aromatic
index represented by Xc based on assigned elemental combinations using eqs. (1) and (2), where $m$
and $k$ were the fractions of oxygen and sulfur atoms in the π-bond structures of a compound (both $m$
and $k$ were presumed to be 0.50 in this work (Yassine et al., 2014)).
$DBE = c - 0.5h + 0.5n + 1$     (1)
$Xc = (3 \times (DBE - m \times o - k \times s) - 2)/(DBE - m \times o - k \times s)$   (if $DBE < (m \times o +$
$k \times s)$  or $Xc < 0$, then $Xc$ was set to 0)     (2)
In eq. (2), Xc is an important indicator of whether aromatic rings exist in a molecule. Studies
have proved that a molecule is considered aromatic if its Xc value exceeds 2.50 (Ma et al., 2022;
Yassine et al., 2014). OSs were selected based on compounds with O/S ≥ 4 and $HSO_4^-$ ($m/z$ 96.96010)
fragments were observed in their corresponding $MS^2$ spectra. Among them, if N number is 1, O/S ≥ 7,
and their $MS^2$ spectra showed $ONO_2^-$ (*m/z* 61.98837) fragment, these OSs were defined as nitrooxy
OSs (NOSs). It should be noted that several CHOS (composed of C, H, O, and S atoms, hereinafter)
and CHONS species were not determined as OSs due to their low-abundance and insufficient to trigger
reliable data-dependent $MS^2$ acquisition, which may lead to an underestimation of total OS mass
concentration.

## 2.3 Classification and Quantification/Semi-quantification of Organosulfates

To ensure the reliability of quantitative analysis and source attribution, this study focuses on OS
species with $C \geq 8$. The exclusion of smaller OSs ($C \leq 7$) is based on challenges in their unambiguous
identification, including co-elution with interfering compounds (Liu et al., 2024), and higher
uncertainty in precursor assignment due to the lack of characteristic "tracer" molecules in laboratory
experiments. Though re-dissolve using pure methanol may not be the ideal solvent for retaining polar,
early-eluting compounds on the reversed-phase column, it provided a consistent solvent for the
analysis of the mid- and non-polar OS species ($C \geq 8$) that are the focus of this study.
To classify the identified OSs, we employed and compared two distinct classification approaches.
Firstly, a conventional classification approach relies primarily on precursor–product relationships
established through controlled laboratory chamber experiments and field campaigns (Zhao et al., 2018;
Wang et al., 2021a; Deng et al., 2021; Xu et al., 2021b; Mutzel et al., 2015; Brüggemann et al., 2020;
Yang et al., 2024; Duporté et al., 2020; Huang et al., 2023b; Wang et al., 2022b; Riva et al., 2016a).
Based on these established precursor–product relationships, detected OSs and NOSs were classified
into four groups: Monoterpene OSs (including Monoterpene NOSs, hereinafter), Aliphatic OSs
(including Aliphatic NOSs, hereinafter), Aromatic OSs (including Aromatic NOSs, hereinafter), and
Sesquiterpene OSs (including Sesquiterpene NOSs, hereinafter) (see Table S4 for details). It is
apparently that this approach has notable limitations when applied to detected OS in atmospheric
aerosols. A substantial fraction of detected OSs does not match known laboratory tracers and are thus
labeled Unknown OSs (including Unknown NOSs, hereinafter).
Synthetic α-pinene OSs ($C_{10}H_{17}O_5S^-$) and NOSs ($C_{10}H_{16}NO_7S^-$) served for (semi-)quantifying
Monoterpene and Sesquiterpene OSs. Their detailed synthesis procedure was described in previous
study (Wang et al., 2019b). Potassium phenyl sulfate ($C_6H_5O_4S^-$) and sodium octyl sulfate ($C_8H_{17}O_4S^-$)
were used for Aromatic OSs and Aliphatic OSs due to lack of authentic standards (Yang et al., 2023;
He et al., 2022; Staudt et al., 2014). Unknown OSs were semi-quantified by surrogates with similar
retention times (RT) (Yang et al., 2023; Huang et al., 2023b). Table 1 lists the standards, retention
times, and quantified categories. Unknown OSs were absent between 2.00–5.00 min and after 13.60
min.
**Table 1** Chemical structure, UHPLC retention time, and quantified categories of standards used in
the quantification/semi-quantificaiton of OSs and NOSs

| Formula (M-H) | *m/z* ([M-H]$^-$) | Chemical structure | UHPLC RT (min) | Quantified OSs categories |
|---|---|---|---|---|
| $C_6H_5O_4S^-$ | 172.99140 |  | 0.92 | Aromatic OSs, Unknown OSs (RT 0.50-2.00 min) |

| Formula | Mass | Structure | RT | Classification |
|---|---|---|---|---|
| $C_8H_{17}O_4S^-$ | 209.08530 |  | 10.30 | Aliphatic OSs, Unknown OSs (RT 10.00-13.60 min) |
| $C_{10}H_{17}O_5S^-$ | 249.08022 |  | 7.73 | Monoterpene OSs, Sesquiterpene OSs, Unknown OSs (RT 5.00-10.00 min) |
| $C_{10}H_{16}NO_7S^-$ | 294.06530 |  | 9.26 | Monoterpene NOSs and Sesquiterpene NOSs |

202       This quantification approach introduces inherent uncertainty, as differences in molecular
structure and functional groups between a surrogate and detected OSs have different ionization
efficiency (Ma et al., 2025), which is a well-documented challenge in NTA of complex mixtures.
However, this approach provides a consistent basis for comparing the relative abundance of OS in
different cities and their formation driving factors. Hence, the mass concentration of detected OSs is
still reliable in understanding their classification and formation driving factors.

208       To classify the Unknown OSs, we first calculated the Xc of each specie. Those with DBE > 2 and
Xc > 2.50 were designated as Aromatic OSs (Yassine et al., 2014). Subsequently, constrained positive
matrix factorization (PMF) analysis was performed using EPA PMF 5.0. The input matrix comprised
the mass concentrations of 60 unclassified OS species across all samples.

212       Figure S2 shows the source profiles of PMF model. Four factors were identified in this study.
Specifically, Factor 1 is identified as Aliphatic OSs due to the dominant contributions from species
like $C_{11}H_{22}O_5S$ and $C_{12}H_{24}O_5S$, which possess low DBE and are characteristic of long-chain alkane
oxidation (Yang et al., 2024). This assignment is strongly supported by the co-variation of this factor
with n-dodecane. Similarly, Factor 2 is classified as Aromatic OSs, highlighted by the significant
contribution of $C_{10}H_{10}O_7S$ and $C_{11}H_{14}O_7S$, which have been proved as OSs derived from typical
aromatic VOCs (Riva et al., 2015). In addition, the high contributions of benzene, toluene, and styrene
in Factor 2 further suggests that this factor should be classified as Aromatic OSs. As for Factor 3 and
Factor 4 is confirmed by the prominence of established Monoterpene OSs (Surratt et al., 2008; Iinuma
et al., 2007) (e.g., $C_{10}H_{18}O_5S$, $C_{10}H_{17}NO_7S$) and Sesquiterpene OSs (Wang et al., 2022b) (e.g.,
$C_{14}H_{28}O_6S$, $C_{15}H_{25}NO_7S$), respectively. Moreover, isoprene showed high contribution in both Factors
3 and 4. As monoterpenes and sesquiterpenes cannot be detected by online GC-MS, considering that
monoterpenes and sesquiterpenes mainly originate from biogenic sources and strongly correlate with
isoprene (Guenther et al., 2006; Sakulyanontvittaya et al., 2008), therefore, isoprene is used as a
surrogate marker as Monoterpene OSs and Sesquiterpene OSs. High contribution of isoprene in
Factors 3 and 4 proved that these factors were respectively determined as Monoterpene OSs and
Sesquiterpene OSs. Based on marker species, Unknown OSs were further categorized into
Monoterpene, Aromatic, Aliphatic, and Sesquiterpene OSs.

230       The model was executed with 10 runs to ensure stability. The ratio of $Q_{robust}/Q_{true}$ for this solution
was stabilized below 1.50, indicating a robust fit without over-factorization. Furthermore, the scaled
residual matrix (see Figure S3), demonstrating that residuals are randomly distributed and
predominantly within the acceptable range of -3 to 3. Correlation coefficients between classified OSs
and corresponding VOCs (Monoterpene OSs vs. isoprene; Aromatic OSs vs. benzene; Aliphatic OSs

vs. n-dodecane; Sesquiterpene OSs vs. isoprene) were calculated as a statistical auxiliary variable to verify the reliability of PMF results. The arithmetic mean of hourly VOCs within each corresponding filter sampling period was calculated to align the time resolution of VOCs and OS mass concentration. Species with $R < 0.40$ were excluded to avoid potential incorrect classification.

To validate classification accuracy, $MS^2$ fragment patterns were analyzed (Table S5). Diagnostic fragments supported the assignments: Aliphatic OSs showed sequential alkyl chain cleavages ($\Delta$ $m/z$ = 14.0157) and saturated alkyl fragments ($[C_nH_{2n+1}]^-$ or $[C_nH_{2n-1}]^-$); Monoterpene OSs displayed $[C_nH_{2n-3}]^-$ fragments; Aromatic OSs exhibited characteristic aromatic substituent fragments ($[C_6H_5R-H]^-$, R = alkyl, carbonyl, -OH, or H). While absolute certainty for every individual OS in a complex ambient mixture is unattainable, integrating the precursor-constrained PMF model, tracer VOCs correlation analysis, and $MS^2$ fragment patterns validation significantly reduces the likelihood of systematic misclassification.

# 3. Results and Discussion

## 3.1 Concentrations, Compositions, and Classification of Organosulfates

Figure 1 shows the temporal variations of OS and organic aerosols (OA) mass concentrations, as well as RH, during the sampling period across the three cities. The total mass concentration of OS reported in this study is the sum of the (semi-)quantified concentrations of all individual OS species that met the identification criteria described in Section 2.3. The mean OSs concentrations were ($41.1 \pm 34.5$) ng/m$^3$ in Beijing, ($57.4 \pm 39.2$) ng/m$^3$ in Taiyuan, and ($102.1 \pm 80.5$) ng/m$^3$ in Changsha. Table S6 summarizes the average concentrations of PM$_{2.5}$, OC, gaseous pollutants, OS mass concentrations, and the mean meteorological parameters during sampling period for all three cities. OS accounted for $0.64\% \pm 0.44\%$, $0.41\% \pm 0.24\%$, and $0.76\% \pm 0.34\%$ of the total OA in Beijing, Taiyuan, and Changsha, respectively.

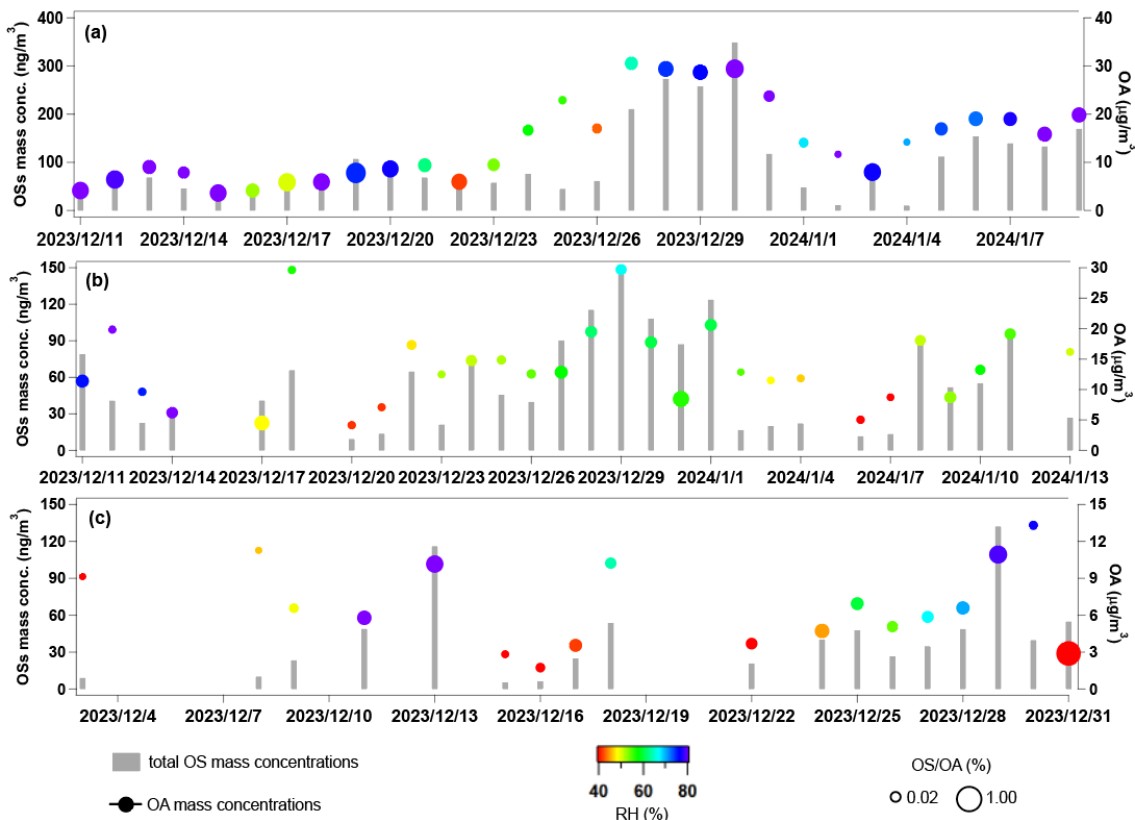

**Figure 1** Temporal variations of daily total OS mass concentrations and average OA mass concentrations in (a) Changsha, (b) Taiyuan, and (c) Beijing. The markers of OA mass concentrations are colored by average RH during sampling period, and marker sizes indicate the OS/OA mass concentration ratios.

The highest OS mass concentrations and OS/OA ratios were observed in Changsha. As shown in Figure 1(a), a distinct episode with OS mass concentrations exceeding 300 ng/m³ occurred between December 27th and 31st, leading to the elevated OS mass concentrations in Changsha. This episode coincided with a period of intense fireworks activity, as evidenced by significant increases in the concentrations of recognized fireworks tracers, especially Ba and K (see Figure S4), leading to an increase in $SO_2$ emission. We noted that though K may originate from biomass burning, its trend in concentration shows good consistency with that of Ba. Therefore, we still infer that fireworks activity are also the primary source of K. Considering persistently high RH (consistently >70%) during this period, as displayed in Figure S5, ALWC (117.9 μg/m³ in average) therefore increased and facilitated the heterogeneous oxidation of $SO_2$ to particulate sulfate (Wang et al., 2016a; Ye et al., 2023). Since particulate sulfate serves as a key reactant in OS formation pathways, its elevated concentration directly promoted OS production (Xu et al., 2024; Wang et al., 2020). Furthermore, fireworks activity led to concurrent increases in the concentrations of transition metals, notably Fe and Mn (Figure S4), which are known to catalyze aqueous-phase radical chemistry and OS formation (Huang et al., 2019; Huang et al., 2018a). Therefore, the pronounced OS mass concentration during this period is attributed to a combination of elevated precursor emissions ($SO_2$), high-RH conditions favoring aqueous-phase processing, and the potential catalytic role of co-emitted transition metals.

It is noteworthy that the single highest OS/OA ratio in Beijing was observed on December 31st under low RH. This phenomena highlights that ALWC, while a major driving factor of OS formation,

is not an exclusive control. Specifically, this day showed high atmospheric oxidative capacity and
aerosol acidity. We note that under such conditions, efficient acid-catalyzed heterogeneous reactions
of gas-phase oxidation products could drive substantial OS formation. The impact of ALWC,
atmospheric oxidative capacity, and aerosol pH on OS formation will be discussed in detail in Section
286 3.2.

Figures 2(a) and 2(b) shows the average mass concentrations and fractions of different OSs
categories across the three cities, based on classification approach based on OSs' elemental
composition and laboratory chamber-derived precursor–OS relationships and our precursor-based
PMF classification approach developed in this work, respectively (see Section 2.3 for details). As
displayed in Figure 2(b), Monoterpene OSs dominated detected OSs across all cities, contributing 55.2%
(Beijing), 46.8% (Taiyuan), and 72.3% (Changsha) to total OS, respectively. Biogenic-emitted
monoterpene is the precursor of Monoterpene OSs. However, monoterpenes are primarily biogenic
precursors, their limited emissions during winter cannot fully explain the high mass fractions of
Monoterpene OSs. Recent studies have highlighted anthropogenic sources, particularly biomass
burning, as significant contributors to monoterpene (Wang et al., 2022a; Koss et al., 2018). The $PM_{2.5}$
source apportionment analysis (Text S2, Figure S6) confirmed that biomass burning substantially
contributed to $PM_{2.5}$ across all cities. The highest total mass fractions of Monoterpene OSs in
Changsha are mainly attributed to the high RH (Table S6), which facilitates their formation via
heterogeneous reactions (Hettiyadura et al., 2017; Wang et al., 2018; Ding et al., 2016a; Ding et al.,
2016b; Li et al., 2020).
In Taiyuan, the total mass fractions of Aromatic OSs (21.2%) were significantly higher than those
in Beijing (10.7%) and Changsha (4.6%). Aromatic OSs primarily formed via aqueous-phase reactions
between S(IV) and aromatic VOCs (Huang et al., 2020). Taiyuan exhibited the highest sulfate mass
concentration among the three cities (Table S6), which promoted the formation of these species.
Additionally, transition metal ions—particularly $Fe^{3+}$—catalyze aqueous-phase formation of Aromatic
OSs (Huang et al., 2020). High Fe mass concentration was observed in Taiyuan ($0.79 \pm 0.53$ μg/m³),
further facilitated the formation of Aromatic OSs.
The highest total mass fractions of Aliphatic OSs were observed in Beijing (28.1%). Since vehicle
emissions, which is an important source of long-chain alkenes (He et al., 2022; Wang et al., 2021a;
Riva et al., 2016b; Tao et al., 2014; Tang et al., 2020), substantially contributed to $PM_{2.5}$ in all cities
(Figure S6), the relative dominance of Aliphatic OSs in Beijing can be attributed to a comparative
reduction in the emissions of precursors for Monoterpene OSs and Aromatic OSs. Specifically, Beijing
exhibits lower emissions of monoterpene and aromatic VOCs precursors relative to Taiyuan and
Changsha, which results in a reduced contribution of Monoterpene and Aromatic OSs to the total OS
(see Figure 2(b)). Therefore, the relative mass fraction of Aliphatic OSs, which primarily derived from
between sulfate and photooxidation products of alkenes (Riva et al., 2016b), becomes more prominent
in Beijing. Additionally, low RH in Beijing further suppresses the aqueous-phase formation of
Monoterpene OSs, amplifying the relative importance of Aliphatic OSs.

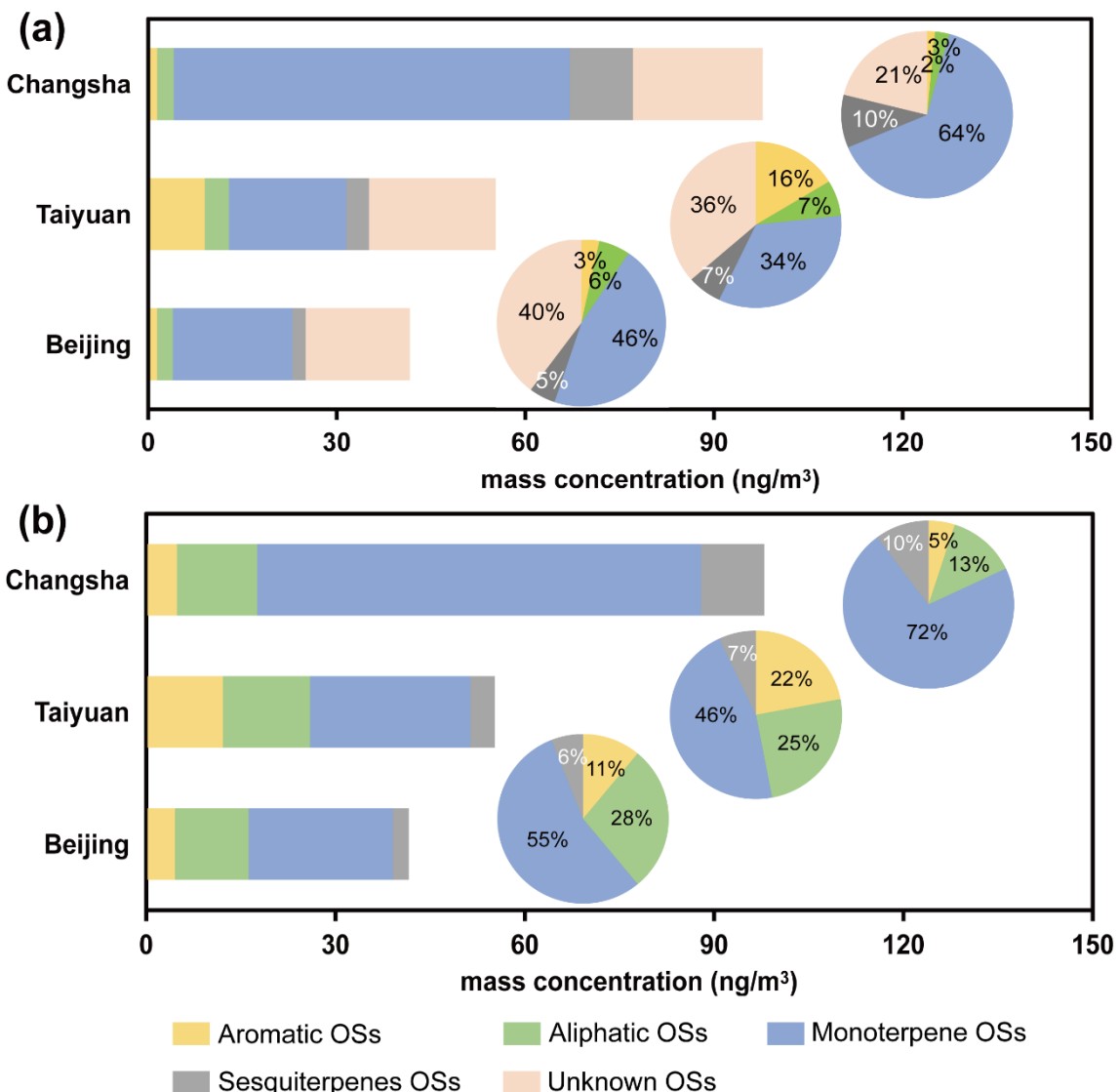

**Figure 2** The average mass concentrations of different OSs categories using (a) conventional classification approach based on OSs' elemental composition and laboratory chamber-derived precursor–OS relationships and (b) precursor-based PMF classification approach developed in this work across all cities. The inserted pie chart indicates the average mass fractions of different OSs categories.

## 3.2 Formation Driving Factors of Aliphatic OSs and NOSs

Compared with conventional classification approach (Figure 2(a)), we found Aliphatic OSs increased markedly (by 22.0%, 17.8%, and 10.3% in Beijing, Taiyuan, and Changsha, respectively). Therefore, we further examined the formation drivers of Aliphatic OSs.

ALWC plays a key role in facilitate OS formation (Wang et al., 2020). Using $PM_{2.5}$ chemical composition and RH, ALWC was calculated via the ISORROPIA-II model (details in Text S3) (Fountoukis and Nenes, 2007). Given the direct influence of ambient RH on ALWC (Figure S7) (Bateman et al., 2014) and leveraging RH-resolved samples from Beijing and Taiyuan, we assessed RH effects on Aliphatic OSs under low (RH < 40%), medium (40% ≤ RH < 60%), and high (RH ≥

60%) conditions.

In Changsha, where RH remains consistently high, Aliphatic OSs mass concentrations strongly correlated with RH (R = 0.78). In Beijing and Taiyuan, correlations increased from low to medium RH (Beijing: 0.53 to 0.82; Taiyuan: 0.38 to 0.77) but declined slightly at higher RH (Beijing: 0.82 to 0.69; Taiyuan: 0.77 to 0.72). The initial correlation rise reflects ALWC-enhanced sulfate-driven heterogeneous OS formation (Wang et al., 2016b; Cheng et al., 2016b), while the decline at elevated RH may due to the increase in ALWC dilutes the concentrations of precursors and intermediates of Aliphatic OSs within the aqueous phase. Therefore, Aliphatic OSs formation were not further promoted, exhibiting the non-linear response of their mass concentrations and ALWC.

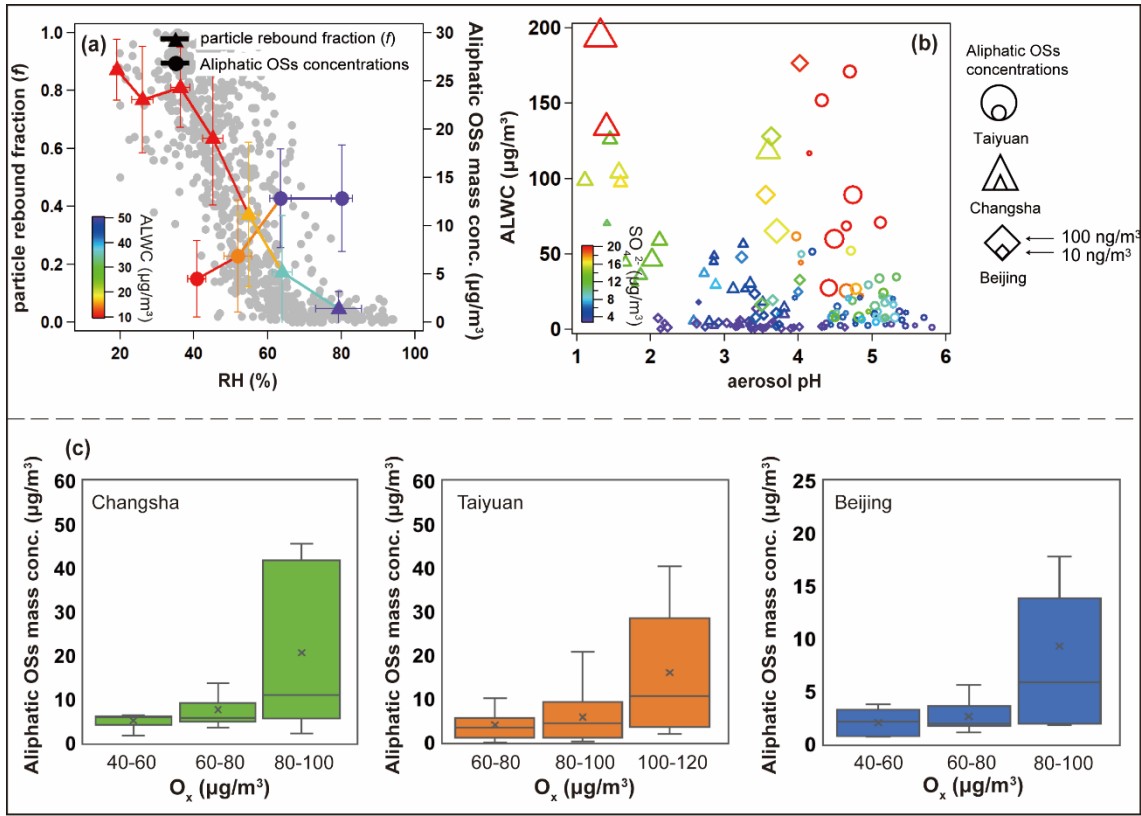

**Figure 3** (a) The measured particle rebound fraction (*f*) and total mass concentrations of Aliphatic OSs as a function of RH, the plots were colored by the calculated ALWC concentrations in Taiyuan, grey dots indicate the mass concentrations of Aliphatic OSs; (b) the relationship between aerosol pH and ALWC across three cities, the markers were colored by the inorganic sulfate mass concentrations, the marker sizes represented the total mass concentrations of Aliphatic OSs; (c) the box plot of total mass concentrations of Aliphatic OSs at different $O_x$ concentration levels.

This threshold behavior aligns with aerosol phase transitions. Particle rebound fraction (*f*), indicating phase state, was measured in Taiyuan using a three-arm impactor (Liu et al., 2017). As RH exceeded 60%, *f* dropped below 0.2 (Figure 3(a)), signaling a transition from non-liquid to liquid aerosol states. This transition at RH > 60% aligns with prior field (Liu et al., 2017; Liu et al., 2023; Meng et al., 2024; Song et al., 2022) and modeling (Qiu et al., 2023) studies in Eastern China. Correspondingly, Aliphatic OSs concentrations increased with RH below 60% but plateaued beyond that despite further humidity rises. These findings underscore aerosol phase state as a critical factor:

initial liquid phase formation (RH < 60%) promotes heterogeneous OS formation (Ye et al., 2018), whereas at higher RH, saturation of reactive interfaces limits further ALWC effects.

In addition, the increase in ALWC with rising RH altered aerosol pH (Figure 3(b)), which inhibited OSs formation via acid-catalyzed reactions (Duporté et al., 2016). In Changsha, as aerosol pH increased from approximately 1.0 to above 3.0, the average total mass concentrations of Aliphatic OSs decreased significantly from 9.3 to 4.6 ng/m$^3$ (Figure S8), with further declines as pH increased. In Taiyuan, OS concentrations dropped from 12.2 to 6.8 ng/m$^3$ as pH rose from below 4.5 to above 5.0. However, in Beijing, total mass concentrations of Aliphatic OSs remained stable within a narrow pH range of 3.2–3.9. Elevated ALWC facilitates aqueous-phase radical chemistry that forms OSs via non-acid pathways, which can dominate over pH-dependent processes (Rudziński et al., 2009; Wach et al., 2019; Huang et al., 2019). Thus, pH-dependent suppression of Aliphatic OSs formation is common across urban aerosol pH ranges, but less evident when pH varies narrowly.

Inorganic sulfate plays a crucial role in OS formation via sulfate esterification reactions (Xu et al., 2024; Wang et al., 2020). We thus examined its effect on the formation of Aliphatic OSs. Figure 3(b) illustrates the relationships among ALWC, pH, inorganic sulfate mass concentration, and total mass concentrations of Aliphatic OSs across all cities. A consistent positive correlation was observed, consistent with previous field studies (Lin et al., 2022; Wang et al., 2023b; Le Breton et al., 2018; Wang et al., 2018). This correlation was strongest when sulfate concentrations were below 20 μg/m$^3$. Below this threshold, total mass concentrations of Aliphatic OSs increased significantly with inorganic sulfate, whereas above it, the correlation weakened. Additionally, inorganic sulfate mass concentration showed a clear positive correlation with ALWC (Figure 3(b)), suggesting that ionic strength did not increase linearly with sulfate mass. This likely reflects saturation effects in acid-mediated pathways, driven by limitations in water activity and ionic strength (Wang et al., 2020). Overall, these results highlight the nonlinear influence of inorganic sulfate on Aliphatic OSs formation.

Atmospheric oxidative capacity, represented by $O_x$ ($O_x = O_3 + NO_2$) concentrations, typically modulates OS formation via acid-catalyzed ring-opening reactions pathways. As shown in Figure 3(c), total mass concentrations of Aliphatic OSs and NOSs exhibited significant increases with rising $O_x$ levels across all cities. Especially, total mass concentrations of Aliphatic OSs significantly increased across all cities when $O_x$ concentrations raised from 60–80 μg/m$^3$ to > 80 μg/m$^3$. As shown in Figure S9, $O_3$ dominated the $O_x$ composition during high-$O_x$ episodes (> 80 μg/m$^3$) across all cities. Previous laboratory studies have suggested that enhanced atmospheric oxidation capacities promote the oxidation of VOCs (Zhang et al., 2022; Wei et al., 2024), forming cyclic intermediates. We therefore inferred that the increase in $O_x$ facilitates the formation of cyclic intermediates derived from long-chain alkenes. Subsequent acid-catalyzed and ring-opening reactions are important pathways of heterogeneous OSs formation, including Aliphatic OSs (Eddingsaas et al., 2010; Iinuma et al., 2007; Brüggemann et al., 2020).

It should be noted that though formation driving factors of Aliphatic OSs identified in this work, including ALWC, inorganic sulfate, and $O_x$, are likely applicable in other urban environments sharing similar winter conditions characterized by high anthropogenic emissions and moderate-to-high humidity. However, their importance may differ in other cities with different atmospheric conditions, like in summer with strong biogenic emissions, in regions with low aerosol acidity, or in arid cities with persistently low RH.

## 4 Conclusions and Implications

In this study, we applied a NTA approach based on UHPLC-HRMS to investigate the molecular composition of OS in $PM_{2.5}$ samples from three cities. By integrating molecular composition data, precursor-constrained PMF source apportionment, and OS–precursor correlation analysis, we developed a comprehensive method for accurate classification of detected OSs, demonstrating superior discrimination between Aliphatic OSs. Conventional classification methods rely on laboratory chamber-derived precursor–OS relationships (Wang et al., 2019a), which provide limited insight into the formation of Aliphatic OSs and tend to underestimate their mass fractions. The abundant Aliphatic OSs detected in ambient $PM_{2.5}$ suggest complex formation pathways, such as OH oxidation of long-chain alkenes (Riva et al., 2016b) and heterogeneous reactions between $SO_2$ and alkene in acidic conditions (Passananti et al., 2016), which remain incompletely understood in laboratory studies. Our findings highlight the importance of emphasizing the formation of Aliphatic OSs in urban atmospheres.

This study still faces several challenges. This work was conducted during the winter. OS formation exhibits seasonal variability, particularly for pathways driven by biogenic VOCs emissions and photochemical activity, which are generally enhanced in warmer months. Hence, the underestimation of Aliphatic OSs, and their key formation factors determined in this work remain valid insights for the winter period but may not fully represent annual OS behavior. In addition, our field campaigns were conducted in three typical different Chinese cites, the effect of these driving factors on the formation of Aliphatic OSs may not be applicable to other cities with different atmospheric conditions. Future long-term observations in more cities are necessary to resolve the complete annual cycle of OS composition, quantify the shifting contributions of anthropogenic versus biogenic precursors, and understanding how key formation driving factors evolve with changing atmospheric conditions.

For NTA, the use of surrogate standards for quantification OS mass concentration introduced uncertainty, particularly due to the extraction efficiency of individual OSs species from quartz fiber filters could not be determined. Although we have adopted standardized extraction protocol ensures high comparability across our samples, absolute extraction recoveries may vary. In addition, this approach depends on public molecular composition such as mzCloud and ChemSpider integrated within the Compound Discoverer software, which contain limited entries for organosulfates. Reliance on these databases for compound identification may therefore underestimate OS mass concentrations in urban environments. For example, OSs identified here accounted for less than 1% of total OA mass, whereas recent work (Ma et al., 2025) reported approximately 20% contributions.

OS may become increasingly significant in OA, particularly in coastal regions influenced by oceanic dimethyl sulfate emissions (Brüggemann et al., 2020). Our future work will focus on synthesizing OSs standards representing various precursors and establishing a dedicated fragmentation database through multi-platform $MS^2$ validation to elucidate OS sources in more detail.

**Author Contributions**

Y.Q., J.W., and Z.W. designed this work. J.L., Y.Wei, C.L., J.Y., T.L., R.M., T.Z., W.F., J.Y., Z.F., Y.X.

and K.B. collected PM$_{2.5}$ samples. Y.Q., J.W., T.Q., Y.B., and D.L. conducted UHPLC-HRMS
experiments. Y.Q., J.W., Z.G., and Y.Wang wrote this manuscript. Z.W., Y.Wang, and M.H. edited this
manuscript. All authors have read and agreed to submit this manuscript. Y.Q. and J.W. contributed
equally to this work.

**Funding**

This work is funded by the National Nature Science Foundation of China (Grants 22221004 and
22306059), This work was also supported by the Science and Technology Innovation Program of
Hunan Province (Grants 2024RC3106 and 2025AQ2001), and the Fundamental Research Funds
for the Central Universities (Grant 531118010830).

**Notes**

The authors declare that they have no conflict of interest.

**Acknowledgements**

Y.W would like to acknowledge financial support by the National Nature Science Foundation of
China (Grants 22221004 and 22306059), This work was also supported by the Science and
Technology Innovation Program of Hunan Province (Grants 2024RC3106 and 2025AQ2001),
and the Fundamental Research Funds for the Central Universities (Grant 531118010830).

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
