# Peer review of "Underestimation of Anthropogenic Organosulfates in Atmospheric"

_EGUsphere, 2025_

## Author Comment (AC1)

**Response to Referee#1**

Thanks to your careful reading and their constructive comments and suggestions on our manuscript. The reviewers' comments and suggestions are shown as *italicized font*, our response to the comments is normal font. New or modified text is in normal font and in blue. Details are as follows.

- - - - - - - - - - - - - - - - - - - - - - - - - - - - - - - - - - - - - - - - - - - - - - - - - -

Referee's comments:

*This study reports the characterization of organosulfates (OSs) in PM2.5 samples collected from three urban sites using UHPLC-HRMS to obtain molecular composition data to perform precursor-constrained PMF source apportionment and OSs–precursor correlation analysis for classifcation of detected OSs.*

[response]

Thanks for your comments. Please check our point-by-point response.

*While this study presents a lot of data, it remains unclear to me how precursor-constrained PMF source apportionment was performed in this study using molecular composition data from UHPLC-HRMS. Specifically, I did not find details about how the source profiles were established and the critical QA/QC elements (e.g., the residual matrix, Q value, and related diagnostics) for validating the statistical robustness and chemical realism of the results. The testing variables and correlation analysis results were not presented in a clear manner, making it difficult to assess the validity of the findings. Substantial revisions and further clarifications are required to help the readers better understand the reported data and its implications.*

[response]

Thanks for your comments on the technical details about the PMF model used in this work. We acknowledge that the initial submission lacked sufficient description of the PMF's setup, validation diagnostics, and the chemical rationale behind the resolved source profiles. We have thoroughly revised the manuscript to provide a comprehensive account of these elements.

Figure R1 shows the source profiles of PMF model. Four factors were identified in this study. Specifically, Factor 1 is identified as Aliphatic OSs (including Aliphatic NOSs, hereinafter) due to the dominant contributions from species like $C_{11}H_{22}O_5S$ and $C_{12}H_{24}O_5S$, which possess low DBE and are characteristic of long-chain alkane oxidation. This assignment is strongly supported by the co-variation of this factor with n-dodecane. Similarly, Factor 2 is classified as Aromatic OSs, highlighted by the significant contribution of $C_{10}H_{10}O_7S$ and $C_{11}H_{14}O_7S$, which have been proved as OSs

derived from 2-methylnaphthalene. In addition, the high contributions of benzene, toluene, and styrene in Factor 2 further suggests that this factor should be classified as Aromatic OSs. As for Factor 3 and Factor 4 is confirmed by the prominence of established monoterpene-derived OSs (e.g., $C_{10}H_{18}O_5S$, $C_{10}H_{17}NO_7S$) and sesquiterpene-derived OSs (e.g., $C_{14}H_{28}O_6S$, $C_{15}H_{25}NO_7S$), respectively. Moreover, isoprene showed high contribution in both Factors 3 and 4. Monoterpenes and sesquiterpenes primarily originate from biogenic sources and strongly correlate with isoprene. Therefore, Factors 3 and 4 were respectively determined as Monoterpene OSs and Sesquiterpene OSs.

[Figure]

**Figure R1** The factor profile of the precursor-constrained PMF model

As for the QA/QC elements, the uncertainty for each input data point was meticulously calculated based on the method detection limit. The number of run was set as 10. We evaluated the model's robustness by analyzing the Q values. The ratio of $Q_{robust}/Q_{true}$ stabilized below 1.50, indicating a stable model without over-fitting. Furthermore, the scaled residual matrix is now provided (Figure R2), demonstrating that residuals are randomly distributed and predominantly within the acceptable range of -3 to 3. These diagnostics confirm the statistical soundness of the solution.

[Figure]

**Figure R2** The scaled residual matrix of precursor-constrained PMF model

We hope these substantial revisions satisfactorily clarify the methodology and enhance the reader's ability to assess the validity and implications of our reported data.

[revised]

Lines 206–236:

To classify the Unknown OSs, we first calculated the Xc of each specie. Those with DBE > 2 and Xc > 2.50 were designated as Aromatic OSs (Yassine et al., 2014). Subsequently, constrained positive matrix factorization (PMF) analysis was performed using EPA PMF 5.0. The input matrix comprised the mass concentrations of 60 unclassified OS species across all samples.

Figure S2 shows the source profiles of PMF model. Four factors were identified in this study. Specifically, Factor 1 is identified as Aliphatic OSs due to the dominant contributions from species like $C_{11}H_{22}O_5S$ and $C_{12}H_{24}O_5S$, which possess low DBE and are characteristic of long-chain alkane oxidation (Yang et al., 2024). This assignment is strongly supported by the co-variation of this factor with n-dodecane. Similarly, Factor 2 is classified as Aromatic OSs, highlighted by the significant contribution of $C_{10}H_{10}O_7S$ and $C_{11}H_{14}O_7S$, which have been proved as OSs derived from typical aromatic VOCs (Riva et al., 2015). In addition, the high contributions of benzene, toluene, and styrene in Factor 2 further suggests that this factor should be classified as Aromatic OSs. As for Factor 3 and Factor 4 is confirmed by the prominence of established Monoterpene OSs (Surratt et al., 2008; Iinuma et al., 2007) (e.g., $C_{10}H_{18}O_5S$, $C_{10}H_{17}NO_7S$) and Sesquiterpene OSs (Wang et al., 2022b) (e.g., $C_{14}H_{28}O_6S$, $C_{15}H_{25}NO_7S$), respectively. Moreover, isoprene showed high contribution in both Factors 3 and 4. As monoterpenes and sesquiterpenes cannot be detected by online GC-MS, considering that monoterpenes and sesquiterpenes mainly originate from biogenic sources and strongly correlate with isoprene (Guenther et al., 2006; Sakulyanontvittaya et al., 2008), therefore, isoprene is used as a surrogate marker as Monoterpene OSs and Sesquiterpene OSs. High contribution of isoprene in Factors 3 and 4 proved that these factors were respectively determined as Monoterpene OSs and Sesquiterpene OSs. Based on marker species, Unknown OSs were further categorized into Monoterpene, Aromatic, Aliphatic, and Sesquiterpene OSs.

The model was executed with 10 runs to ensure stability. The ratio of $Q_{robust}/Q_{true}$ for this solution was stabilized below 1.50, indicating a robust fit without over-factorization. Furthermore, the scaled residual matrix (see Figure S3), demonstrating that residuals are randomly distributed and predominantly within the acceptable range of -3 to 3. Correlation coefficients between classified OSs and corresponding VOCs (Monoterpene OSs vs. isoprene; Aromatic OSs vs. benzene; Aliphatic OSs vs. n-dodecane; Sesquiterpene OSs vs. isoprene) were calculated as a statistical auxiliary variable to verify the reliability of PMF results. The arithmetic mean of hourly VOCs within each corresponding filter sampling period was calculated to align the time resolution of VOCs and OS mass concentration. Species with R < 0.40 were excluded to avoid potential incorrect classification.

Figure S2:

Same as Figure R1.

Figure S3:

Same as Figure R2.

*Specific Comments*

*(1) Lines 92-97: Can the authors provide some scientific rationales for why these three sampling sites were selected? Do they have distinct emission sources or meteorological*

*conditions that can be tested as variables?*

[response]

Thank you for your comment. The selection of the three sampling cities, i.e., Beijing, Taiyuan, and Changsha, was deliberate, based on their representative differences in meteorological conditions and primary emission profiles during winter.

From a meteorological perspective, Beijing and Taiyuan in northern China experience cold, dry winters with low RH. In contrast, Changsha, locates in the humid subtropical region of southern China, is characterized by relatively higher winter RH. This pronounced north-south humidity gradient allows us to explicitly investigate the role of ALWC, which is a critical factor in aqueous-phase OS formation (Cheng et al., 2016; Wu et al., 2018; Zheng et al., 2015).

Regarding the sources of $PM_{2.5}$, the three cities represent distinct dominant $PM_{2.5}$ source regimes. Coal combustion and industrial are two important $PM_{2.5}$ sources in Taiyuan, which is a traditional industrial and coal-mining base. Changsha's $PM_{2.5}$ profile is more dominated by vehicle emissions and domestic cooking. Beijing presents a complex mix where secondary formation processes become the predominant source of $PM_{2.5}$. Figure S4 in Supplementary Information shows the PMF results of $PM_{2.5}$ across three cities. These differences in primary emission types lead to varying abundances of key OS precursors, such as VOCs species and $SO_2$.

In summary, both meteorological conditions (particularly RH) and precursor emission profiles are major drivers of spatial heterogeneity in OS molecular composition and concentration. Therefore, selecting these three cities is suitable to investigate how these key variables govern OSs formation mechanisms across diverse urban environments in China.

[revised]

Lines 93–104:

The site selection was based on contrasts in winter meteorological conditions and dominate $PM_{2.5}$ sources. For meteorological conditions, Beijing and Taiyuan represent northern Chinese cities with cold, dry conditions (low RH). In comparison, Changsha is characterized by relatively higher winter RH. In terms of $PM_{2.5}$ sources, Taiyuan is a traditional industrial and coal-mining base, Changsha's pollution profile is more influenced by traffic and domestic cooking emissions, whereas Beijing is characterized by a high mass fraction of secondary aerosols. This enables a comparative analysis of OS formation mechanisms under varied atmospheric conditions. In Beijing, $PM_{2.5}$ samples were collected at the Peking University Atmosphere Environment Monitoring Station (PKUERS; 40.00°N, 116.32°E), as detailed in previous studies (Wang et al., 2023a). Sampling in Taiyuan and Changsha took place on rooftops at the Taoyuan National Control Station for Ambient

Air Quality (37.88°N, 112.55°E) and the Hunan Hybrid Rice Research Center (28.20°N, 113.09°E), respectively (see Figure S1).

*(2) Lines 111-114: Did the authors have estimates for the extraction efficiencies of OSs from filters using this protocol? This information is critical to assess analytical uncertainties and validate quantification.*

[response]

Thanks for your comment. Thank you for raising this critical point regarding extraction efficiency. We fully agree that this information is important for a complete uncertainty assessment. However, due to the fundamental characteristics of our non-target analysis (NTA) workflow, determining extraction efficiencies for individual OS presents a significant challenge.

In the NTA approach, the molecular composition and concentration of OS are unknown prior to HPLC-HRMS analysis and subsequent data processing. Quantitative estimation relies on surrogate standards (Ma et al., 2022; Qiu et al., 2024), as authentic standards for the vast majority of detected OSs are unavailable. While we used specific surrogates for quantification of different OSs classifications, obtaining extraction efficiencies would require separate experiments with each corresponding authentic compound, which is currently impractical given the structural diversity encountered.

This inherent limitation of semi-quantification in NTA is acknowledged in the field and contributes to the overall analytical uncertainty (Nozière et al., 2015). To address this limitation, we have added relevant discussion in the revised manuscript. Our study's comparative conclusions on OS abundance and formation drivers across cities remain robust, as all samples were processed using the identical, rigorously controlled extraction and analysis approach.

[revised]

Lines 413–421:

For NTA, the use of surrogate standards for quantification OS mass concentration introduced uncertainty, particularly due to the extraction efficiency of individual OSs species from quartz fiber filters could not be determined. Although we have adopted standardized extraction protocol ensures high comparability across our samples, absolute extraction recoveries may vary. In addition, this approach depends on public molecular composition such as mzCloud and ChemSpider integrated within the Compound Discoverer software, which contain limited entries for organosulfates. Reliance on these databases for compound identification may therefore underestimate OS mass concentrations in urban environments. For example, OSs identified here accounted for less than 1% of total OA mass, whereas recent work (Ma et al., 2025) reported approximately 20% contributions.

*(3) Lines 138-143: Can the authors provide some context about what "Xc" means and how this value can aid in the identification of OSs?*

[response]

    Thanks for your comment. Xc is an important indicator of whether aromatic rings exist in a molecule. Xc is calculated from the molecular formula derived from HRMS data, as detailed in our revised manuscript (eq. (2)). The established interpretation is that a molecule is considered aromatic if its Xc value exceeds 2.50. This threshold is widely used in NTA of complex organic mixtures, including atmospheric aerosols, to rapidly filter and classify compounds. In this work, OSs with Xc above 2.50 were classified into Aromatic OSs (including Aromatic NOSs). This provided an efficient, formula-based method to distinguish aromatic-derived OSs from their aliphatic or other counterparts. We will add a brief explanatory sentence in the revised manuscript.

[revised]
Lines 154–169:

    We calculated the double bond equivalent (DRE) and aromatic index represented by Xc based on assigned elemental combinations using eqs. (1) and (2), where m and k were the fractions of oxygen and sulfur atoms in the $\pi$-bond structures of a compound (both m and k were presumed to be 0.50 in this work (Yassine et al., 2014)).

$$DBE = c - 0.5h + 0.5n + 1 \qquad (1)$$

$$Xc = (3 \times (DBE - m \times o - k \times s) - 2)/(DBE - m \times o - k \times s) \quad (if\ DBE < (m \times o + k \times s)\ or\ Xc < 0, then\ Xc\ was\ set\ to\ 0) \qquad (2)$$

    In eq. (2), Xc is an important indicator of whether aromatic rings exist in a molecule. Studies have proved that a molecule is considered aromatic if its Xc value exceeds 2.50 (Ma et al., 2022; Yassine et al., 2014). OSs were selected based on compounds with O/S $\geq$ 4 and $HSO_4^-$ (*m/z* 96.96010) fragments were observed in their corresponding $MS^2$ spectra. Among them, if N number is 1, O/S $\geq$ 7, and their $MS^2$ spectra showed $ONO_2^-$ (*m/z* 61.98837) fragment, these OSs were defined as nitrooxy OSs (NOSs). It should be noted that several CHOS (composed of C, H, O, and S atoms, hereinafter) and CHONS species were not determined as OSs due to their low-abundance and insufficient to trigger reliable data-dependent $MS^2$ acquisition, which may lead to an underestimation of total OS mass concentration.

*(4) Line 151: Can the authors provide more detailed information about what "conventional classification approach" means and specify how it is different from the current precursor-constrained PMF source apportionment?*

[response]

Thank you for your comment. The "traditional classification methods" referred to in our study mainly denote the widely used framework in the literature for the identification and classification of OS, which is based on laboratory experiments and MS characteristics.

(1) **Precursor-controlled laboratory chamber and/or flow tube experiments.** In laboratory chamber or flow tube experiments, only specified target VOCs, such as isoprene, α-pinene, β-caryophyllene, toluene, and etc. are introduced, together with seed particles like ammonium sulfate under the oxidation of $O_3$ and OH radicals, so that the OS formed in the resulting SOA can be identified in a relatively simple system. For example, Surratt et al. used biogenic SOA smog chamber experiments to provide the first clear evidence that the isoprene oxidation product IEPOX can further react with acidic sulfate particles to form typical isoprene-derived OSs such as $C_5H_{12}O_7S$ (Froyd et al., 2010; Surratt et al., 2007; Martinsson et al., 2017). Similar approach has also been applied to α-pinene and other monoterpene as well as sesquiterpene precursors, leading to the identification of a series of $C_{10}$ and $C_{15}$ OSs/NOSs species (Surratt et al., 2007; Hatch et al., 2011). From these experiments, several precursor-specific OSs are determined.

(2) **Establishing "tracer OS–precursor VOC" lists.** On the basis of the laboratory experiments and OS-precursor relationship proved by previous studies, a number of reviews and regional studies in recent years have systematically compiled those "tracer-type OS" that have been repeatedly confirmed, thereby forming OS lists categorized by precursor VOCs type. Table S4 in the Supplementary Information summarizes the OS-precursor relationship. For instance, $C_5H_{12}O_7S$ and $C_5H_8O_7S$ are widely accepted as representative OS formed via the high- and low-NOx pathways of isoprene/IEPOX, respectively; a suite of $C_{10}H_{18}O_5S$ and $C_{10}H_{16}NO_7S$ species are classified as monoterpene-derived OS/NOS; and some $C_2$–$C_3$ small-molecule OS (e.g., glycolic acid sulfate) are generally regarded as arising from mixed contributions of small carbonyls and carboxylic acids (Brüggemann et al., 2020; Fan et al., 2022; Tsona Tchinda et al., 2025).

However, OS classification methods based on the above framework still have notable limitations. The first limitation is a large number of OSs detected in the real atmosphere do not have counterparts in existing lists, leading to a substantial fraction of "Unknown OSs". The second one is relying solely on a small set of parameters such as DBE and H/C typically only allows a coarse separation between broad aliphatic and aromatic classes, making it difficult to achieve a more refined source classification for structurally complex OS. Hence, there is a crucial need for classifying these "Unknown

OSs". We analyze them from multiple perspectives, including molecular-composition metrics, precursor based-PMF source apportionment, and correlation analysis between OS and their putative precursors.

In summary, compared with traditional "precursor-specific OS" classification approach, our classification approach realized accurate classification of detected OSs.

[revised]
Lines 178–189:

To classify the identified OSs, we employed and compared two distinct classification approaches. Firstly, a conventional classification approach relies primarily on precursor–product relationships established through controlled laboratory chamber experiments and field campaigns (Zhao et al., 2018; Wang et al., 2021; Deng et al., 2021; Xu et al., 2021; Mutzel et al., 2015; Brüggemann et al., 2020; Yang et al., 2024; Duporté et al., 2020; Huang et al., 2023b; Wang et al., 2022b; Riva et al., 2016a). Based on these established precursor–product relationships, detected OSs and NOSs were classified into four groups: Monoterpene OSs (including Monoterpene NOSs, hereinafter), Aliphatic OSs (including Aliphatic NOSs, hereinafter), Aromatic OSs (including Aromatic NOSs, hereinafter), and Sesquiterpene OSs (including Sesquiterpene NOSs, hereinafter) (see Table S4 for details). It is apparently that this approach has notable limitations when applied to detected OS in atmospheric aerosols. A substantial fraction of detected OSs does not match known laboratory tracers and are thus labeled Unknown OSs (including Unknown NOSs, hereinafter).

*(5) Line 176: Can the authors explain why isoprene VOC markers were included, given that only OSs with C ≥ 8 were analyzed?*

[response]

Thanks for your comment. The primary reason is a technical limitation in our online VOCs measurement system. While our PMF model aimed to resolve OSs with C atom numbers above 8, our deployed online GC-MS system was not configured to speciate and quantify monoterpenes or sesquiterpenes directly during the campaign. To constrain the biogenic source factors in the model, we therefore needed suitable surrogates. As a major biogenic VOC co-emitted with monoterpenes and sesquiterpenes from biogenic emission, isoprene exhibits strong positive correlation with monoterpenes and sesquiterpenes in ambient air (Guenther et al., 2006; Sakulyanontvittaya et al., 2008). Therefore, we selected isoprene as the auxiliary marker of monoterpene-derived OSs and sesquiterpene-derived OSs in the PMF model. We recognize that direct measurement of monoterpenes and sesquiterpenes would be ideal. However, within the constraints of our observational instrument, the use of isoprene as a correlated biogenic marker represents a scientifically valid and practical

approach to inform the source apportionment. We have now added relevant discussion in Methodology section.

[revised]

Lines 220–227:

Moreover, isoprene showed high contribution in both Factors 3 and 4. As monoterpenes and sesquiterpenes cannot be detected by online GC-MS, considering that monoterpenes and sesquiterpenes mainly originate from biogenic sources and strongly correlate with isoprene (Guenther et al., 2006; Sakulyanontvittaya et al., 2008), therefore, isoprene is used as a surrogate marker as Monoterpene OSs and Sesquiterpene OSs. High contribution of isoprene in Factors 3 and 4 proved that these factors were respectively determined as Monoterpene OSs and Sesquiterpene OSs. Based on marker species, Unknown OSs were further categorized into Monoterpene, Aromatic, Aliphatic, and Sesquiterpene OSs.

*(6) Lines 180-185: For the calculated correlation coefficients, I have a few questions about this section.*

*How were the VOCs collected and analyzed at the sampling site? Were they measured in real time?*

*How were the correlations between classified OSs/NOSs and corresponding VOCs calculated if data were not collected and analyzed at the same time resolution?*

*How did the authors confirm that the measured isoprene emissions were from biogenic sources in winter at the sampling sites, given that isoprene is primarily emitted from deciduous plants?*

[response]

Thanks for your comments.

(1) For the first question, we confirm that ambient VOCs were measured in real-time at each site using an online GC-MS system. This instrument provided hourly-resolved concentration data for a suite of 98 VOCs (see Table R1). Details on the instrument model and analytical procedures are available in our previous study (Qiu et al., 2023).

Table R1 98 VOCs detected by online GC-MS system

| Number | VOCs (ppb) | Number | VOCs (ppb) |
|--------|------------|--------|------------|
| 1 | Ethane | 2 | Propane |
| 3 | Ethylene | 4 | Propylene |
| 5 | Isobutane | 6 | Acetylene |

| | | | |
|---|---|---|---|
| 7 | n-Butane | 8 | Trans-2-butene |
| 9 | 1-Butene | 10 | Cyclopentane |
| 11 | Cis-2-butene | 12 | Isopentane |
| 13 | n-Pentane | 14 | Chloromethane |
| 15 | Freon114 | 16 | Vinylchloride |
| 17 | 1,3-Butadiene | 18 | Bromomethane |
| 19 | Acetaldehyde | 20 | Chloroethane |
| 21 | Freon11 | 22 | trans-2-Pentene |
| 23 | 1-Pentene | 24 | Isoprene |
| 25 | Acrolein | 26 | Propanal |
| 27 | cis-2-Pentene | 28 | 1,1-Dichloroethene |
| 29 | Freon113 | 30 | Acetone |
| 31 | 2,2-Dimethylbutane | 32 | Acetonitrile |
| 33 | Dichloromethane | 34 | 2-Methylpentane |
| 35 | 2,3-Dimethylbutane | 36 | 3-Methylpentane |
| 37 | MTBE | 38 | n-Hexane |
| 39 | 1-Hexene | 40 | Methacrolein |
| 41 | 1,1-Dichloroethane | 42 | n-Butanal |
| 43 | 2,4-Dimethylpentane | 44 | methylvinylketone |
| 45 | Methylcyclopentane | 46 | methylethylketone |
| 47 | cis-1,2-Dichloroethene | 48 | Chloroform |
| 49 | 1,1,1-Trichloroethane | 50 | Cyclohexane |
| 51 | 2-Methylhexane | 52 | 2,3-Dimethylpentane |
| 53 | 3-Methylhexane | 54 | Benzene |
| 55 | Carbontetrachloride | 56 | 2,2,4-Trimethylpentane |
| 57 | 1,2-Dichloroethane | 58 | Trichloroethylene |
| 59 | n-Heptane | 60 | Methylcyclohexane |
| 61 | 2-Pentanone | 62 | n-pentanal |
| 63 | 1,2-Dichloropropane | 64 | 3-pentanone |
| 65 | Bromodichloromethane | 66 | 2-Methylheptane |
| 67 | 2,3,4-Trimethylpentane | 68 | 3-Methylheptane |
| 69 | trans-1,3-Dichloropropene | 70 | n-octane |
| 71 | Toluene | 72 | cis-1,3-Dichloropropene |
| 73 | 1,1,2-trichloroethane | 74 | n-Hexanal |
| 75 | Tetrachloroethylene | 76 | 1,2-Dibromoethane |
| 77 | chlorobenzene | 78 | n-Nonane |
| 79 | Ethylbenzene | 80 | m/p-Xylene |
| 81 | o-Xylene | 82 | iso-Propylbenzene |
| 83 | Styrene | 84 | n-Propylbenzene |
| 85 | m-ethyltoluene | 86 | n-Decane |
| 87 | p-ethyltoluene | 88 | 1,3,5-Trimethylbenzene |
| 89 | o-ethyltoluene | 90 | 1,3-Dichlorobenzene |
| 91 | 1,2,4-Trimethylbenzene | 92 | 1,4-Dichlorobenzene |
| 93 | 1,2,3-Trimethylbenzene | 94 | m-diethylbenzene |

**R11**

| 95 | Benzylchloride | 96 | p-diethylbenzene |
| 97 | 1,2-Dichlorobenzene | 98 | n-undecane |

(2) For the second question, in order to enable a meaningful correlation analysis, we processed the VOCs data by calculating the arithmetic mean of all hourly concentrations measured within each corresponding filter sampling period. This yielded a representative daily average VOCs concentration for each filter, and the reported Pearson's correlation coefficients ($R$) are based on these averaged VOCs values and the mass concentrations of OS. This approach is a standard method for integrating datasets of different frequencies in atmospheric chemistry.

(3) For the third question, we fully acknowledge that biogenic emissions are low in wintertime, and the concentration of isoprene was accordingly low. Within our PMF analysis, isoprene was employed not as an absolute sole tracer but as a statistical auxiliary variable. The rationale was that it would exhibit significant co-variation with the broader biogenic processes contributing to the monoterpene- and sesquiterpene-derived OSs we aimed to resolve. The PMF results support this use, as the relevant biogenic OS factor showed the strongest correlation with isoprene and weak correlations with anthropogenic tracers. We agree that direct measurement of terpenoids would be ideal. We have added more details in our revised manuscript.

[revised]

Lines 231–236:

Correlation coefficients between classified OSs and corresponding VOCs (Monoterpene OSs vs. isoprene; Aromatic OSs vs. benzene; Aliphatic OSs vs. n-dodecane; Sesquiterpene OSs vs. isoprene) were calculated as a statistical auxiliary variable to verify the reliability of PMF results. The arithmetic mean of hourly VOCs within each corresponding filter sampling period was calculated to align the time resolution of VOCs and OS mass concentration. Species with R < 0.40 were excluded to avoid potential incorrect classification.

*(7) Lines 238-248: This paragraph is confusing, and the statement seems contradictory. Can the author clarify how relatively low anthropogenic emissions and low RH promote the dominance of Aliphatic OSs and NOSs, since the precursors for Aliphatic OSs and NOSs (e.g., long-chain alkenes) were also from anthropogenic emissions?*

[response]

We are sorry for the lack of clarity in our original manuscript. We did not intend to suggest that Aliphatic OSs and NOSs are formed under conditions of absolutely low anthropogenic emissions. The point we wanted to convey is based on a relative comparison among the three cities.

Our data indicates that Beijing experiences a different anthropogenic emission profile compared to Taiyuan and Changsha. Specifically, the emissions of precursors critical for forming monoterpene-derived and aromatic OSs and NOSs are relatively lower in Beijing. This results in a lower absolute and relative contribution of Monoterpene OSs/NOSs and Aromatic OSs/NOSs to the total OS budget in Beijing, as shown in Figure 2 in the main text. Therefore, the relative mass fraction of Aliphatic OSs and NOSs, which are primarily generated from the oxidation of long-chain alkanes (e.g., from vehicle emissions) and $SO_2$, becomes more prominent or "dominant" in Beijing's total OS. This prominence is further amplified by the city's typically lower RH, which inhibits the aqueous phase formation of Monoterpene OSs/NOSs. We have revised the relevant paragraph in the manuscript to clarify this comparative context in the revised manuscript.

[revised]
Lines 307–317:

The highest total mass fractions of Aliphatic OSs were observed in Beijing (28.1%). Since vehicle emissions, which is an important source of long-chain alkenes (He et al., 2022; Wang et al., 2021; Riva et al., 2016b; Tao et al., 2014; Tang et al., 2020), substantially contributed to $PM_{2.5}$ in all cities (Figure S4), the relative dominance of Aliphatic OSs in Beijing can be attributed to a comparative reduction in the emissions of precursors for Monoterpene OSs and Aromatic OSs. Specifically, Beijing exhibits lower emissions of monoterpene and aromatic VOCs precursors relative to Taiyuan and Changsha, which results in a reduced contribution of Monoterpene and Aromatic OSs to the total OS (see Figure 2(b)). Therefore, the relative mass fraction of Aliphatic OSs, which primarily derived from between sulfate and photooxidation products of alkenes (Riva et al., 2016b), becomes more prominent in Beijing. Additionally, low RH in Beijing further suppresses the aqueous-phase formation of Monoterpene OSs, amplifying the relative importance of Aliphatic OSs.

*(8) Lines 249-252 (Figure 2): The authors should provide more details about how "classification" was performed.*

[response]
Thanks for your comment. Figure 2(a) illustrates the results obtained using the conventional classification approach, which relies primarily on elemental composition (e.g., carbon number, double bond equivalent) and laboratory chamber-derived precursor–OS relationships. Figure 2(b) presents the results of the precursor-based PMF classification approach developed in this study.

[revised]

**Figure 2** The average mass concentrations of different OSs categories using (a) conventional classification approach based on OSs' elemental composition and laboratory chamber-derived precursor–OS relationships and (b) precursor-based PMF classification approach developed in this work across all cities. The inserted pie chart indicates the average mass fractions of different OSs categories.

*(9) Line 254: The conventional classification approach should refer to Figure 2(a).*

[response]
Thanks for pointing out this typo. We have revised accordingly.

[revised]
Revised accordingly.

*(10) Lines 309-318: This discussion seems mostly speculative. Can the author provide more definitive evidence about how oxidation of long-chain alkenes can potentially form cyclic intermediates that can undergo acid-catalyzed ring-opening pathways for Aliphatic OSs and NOSs formation?*

[response]
Thanks for your comment on the formation mechanism of Aliphatic OSs/NOSs. We acknowledge that the discussion regarding the formation of cyclic intermediates and subsequent acid-catalyzed ring-opening pathways for Aliphatic OSs and NOSs is inferential. However, these discussion is based on mechanistic insights from previous laboratory studies. The offline measurement techniques to directly identify these chemically unstable, short-lived intermediates such as epoxides or Criegee intermediates in ambient aerosol samples is indeed challenging or nearly impossible. The aim of this paragraph is to explore how atmospheric oxidation capacity (represented by Ox concentration) impact the formation of Aliphatic OSS/NOSs, and logically connect field observational evidence to the laboratory-supported chemical mechanisms.

From a mechanistic perspective, laboratory studies have clearly demonstrated that enhanced atmospheric oxidation capacity promotes the formation of cyclic intermediates, and that subsequent acid-catalyzed ring-opening is a crucial pathway for OS formation (Eddingsaas et al., 2010; Iinuma et al., 2007; Riva et al., 2016). Our observational results correlations are consistent with and supportive of such a pathway operating in the urban atmosphere. In the revised manuscript, we have added more discussion and emphasize that this remains an important open question for future

laboratory and targeted field studies.

[revised]
Lines 385–391:

Previous laboratory studies have suggested that enhanced atmospheric oxidation capacities promote the oxidation of VOCs (Zhang et al., 2022; Wei et al., 2024), forming cyclic intermediates. We therefore inferred that the increase in Ox facilitates the formation of cyclic intermediates derived from long-chain alkenes. Subsequent acid-catalyzed and ring-opening reactions are important pathways of heterogeneous OSs formation, including Aliphatic OSs (Eddingsaas et al., 2010; Iinuma et al., 2007; Brüggemann et al., 2020).

*(11) Line 333: What do the "public molecular composition datasets" refer to here?*

[response]

Thank you for your question, and we apologize for the lack of clarity in our manuscript. The term "public molecular composition datasets" refers here to the databases integrated within the Compound Discoverer software (version 3.3, Thermo Scientific) used in our NTA. These include widely referenced spectral and compound libraries such as mzCloud and ChemSpider. In practice, these datasets allow measured high-resolution MS and $MS^2$ spectra to be matched against known reference spectra and molecular structures, thereby assisting in the tentative identification of compounds when authentic standards are unavailable. Relying on these public databases, the mass concentration of OS may be underestimated. We will clarify this terminology in the revised manuscript.

[revised]
Lines 416–421:

In addition, this approach depends on public molecular composition such as mzCloud and ChemSpider integrated within the Compound Discoverer software, which contain limited entries for organosulfates. Reliance on these databases for compound identification may therefore underestimate OS mass concentrations in urban environments. For example, OSs identified here accounted for less than 1% of total OA mass, whereas recent work (Ma et al., 2025) reported approximately 20% contributions.

**References**

Brüggemann, M., Xu, R., Tilgner, A., Kwong, K. C., Mutzel, A., Poon, H. Y., Otto, T., Schaefer, T., Poulain, L., Chan, M. N., Herrmann, H.: Organosulfates in Ambient Aerosol: State of Knowledge and Future Research Directions on Formation, Abundance, Fate, and Importance. Environmental Science & Technology. 54, 3767-3782. 10.1021/acs.est.9b06751, 2020.

Cheng, Y., Zheng, G., Wei, C., Mu, Q., Zheng, B., Wang, Z., Gao, M., Zhang, Q., He, K., Carmichael, G., Pöschl, U., Su, H.: Reactive nitrogen chemistry in aerosol water as a source of sulfate during haze events in China. Science Advances. 2, e1601530. 10.1126/sciadv.1601530, 2016.

Eddingsaas, N. C., VanderVelde, D. G., Wennberg, P. O.: Kinetics and Products of the Acid-Catalyzed Ring-Opening of Atmospherically Relevant Butyl Epoxy Alcohols. The Journal of Physical Chemistry A. 114, 8106-8113. 10.1021/jp103907c, 2010.

Fan, W., Chen, T., Zhu, Z., Zhang, H., Qiu, Y., Yin, D.: A review of secondary organic aerosols formation focusing on organosulfates and organic nitrates. Journal of Hazardous Materials. 430, 128406. https://doi.org/10.1016/j.jhazmat.2022.128406, 2022.

Froyd, K. D., Murphy, S. M., Murphy, D. M., de Gouw, J. A., Eddingsaas, N. C., Wennberg, P. O.: Contribution of isoprene-derived organosulfates to free tropospheric aerosol mass. Proceedings of the National Academy of Sciences. 107, 21360-21365. 10.1073/pnas.1012561107, 2010.

Guenther, A., Karl, T., Harley, P., Wiedinmyer, C., Palmer, P. I., Geron, C.: Estimates of global terrestrial isoprene emissions using MEGAN (Model of Emissions of Gases and Aerosols from Nature). Atmos. Chem. Phys. 6, 3181-3210. 10.5194/acp-6-3181-2006, 2006.

Hatch, L. E., Creamean, J. M., Ault, A. P., Surratt, J. D., Chan, M. N., Seinfeld, J. H., Edgerton, E. S., Su, Y., Prather, K. A.: Measurements of Isoprene-Derived Organosulfates in Ambient Aerosols by Aerosol Time-of-Flight Mass Spectrometry - Part 1: Single Particle Atmospheric Observations in Atlanta. Environmental Science & Technology. 45, 5105-5111. 10.1021/es103944a, 2011.

Iinuma, Y., Müller, C., Berndt, T., Böge, O., Claeys, M., Herrmann, H.: Evidence for the Existence of Organosulfates from β-Pinene Ozonolysis in Ambient Secondary Organic Aerosol. Environmental Science & Technology. 41, 6678-6683. 10.1021/es070938t, 2007.

Ma, J., Ungeheuer, F., Zheng, F., Du, W., Wang, Y., Cai, J., Zhou, Y., Yan, C., Liu, Y., Kulmala, M., Daellenbach, K. R., Vogel, A. L.: Nontarget Screening Exhibits a Seasonal Cycle of PM2.5 Organic Aerosol Composition in Beijing. Environmental Science & Technology. 56, 7017-7028. 10.1021/acs.est.1c06905, 2022.

Martinsson, J., Monteil, G., Sporre, M. K., Kaldal Hansen, A. M., Kristensson, A., Eriksson Stenström, K., Swietlicki, E., Glasius, M.: Exploring sources of biogenic secondary organic aerosol compounds using chemical analysis and the FLEXPART model. Atmos. Chem. Phys. 17, 11025-11040. 10.5194/acp-17-11025-2017, 2017.

Nozière, B., Kalberer, M., Claeys, M., Allan, J., D'Anna, B., Decesari, S., Finessi, E., Glasius, M., Grgić, I., Hamilton, J. F., Hoffmann, T., Iinuma, Y., Jaoui, M., Kahnt, A., Kampf, C. J., Kourtchev, I., Maenhaut, W., Marsden, N., Saarikoski, S., Schnelle-Kreis, J., Surratt, J. D., Szidat, S., Szmigielski, R., Wisthaler, A.: The Molecular Identification of Organic Compounds in the Atmosphere: State of the Art and Challenges. Chemical Reviews. 115, 3919-3983. 10.1021/cr5003485, 2015.

Qiu, Y., Wu, Z., Man, R., Zong, T., Liu, Y., Meng, X., Chen, J., Chen, S., Yang, S., Yuan, B., Song, M., Kim, C., Ahn, J., Zeng, L., Lee, J., Hu, M.: Secondary aerosol formation drives atmospheric particulate matter pollution over megacities (Beijing and Seoul) in East Asia. Atmospheric Environment. 301, 119702. https://doi.org/10.1016/j.atmosenv.2023.119702, 2023.

Qiu, Y., Qiu, T., Wu, Z., Liu, Y., Fang, W., Man, R., Liu, Y., Wang, J., Meng, X., Chen, J., Liang, D., Guo, S., Hu, M.: Observational Evidence of Brown Carbon Photobleaching in Urban Atmosphere at Molecular Level. Environmental Science and Technology Letters. 11, 1032-1039. 10.1021/acs.estlett.4c00647, 2024.

Riva, M., Da Silva Barbosa, T., Lin, Y. H., Stone, E. A., Gold, A., Surratt, J. D.: Chemical characterization of organosulfates in secondary organic aerosol derived from the photooxidation of alkanes. Atmospheric Chemistry and Physics. 16, 11001-11018. 10.5194/acp-16-11001-2016, 2016.

Sakulyanontvittaya, T., Guenther, A., Helmig, D., Milford, J., Wiedinmyer, C.: Secondary Organic Aerosol from Sesquiterpene and Monoterpene Emissions in the United States. Environmental Science & Technology. 42, 8784-8790. 10.1021/es800817r, 2008.

Surratt, J. D., Kroll, J. H., Kleindienst, T. E., Edney, E. O., Claeys, M., Sorooshian, A., Ng, N. L., Offenberg, J. H., Lewandowski, M., Jaoui, M., Flagan, R. C., Seinfeld, J. H.: Evidence for Organosulfates in Secondary Organic Aerosol. Environmental Science & Technology. 41, 517-527. 10.1021/es062081q, 2007.

Tsona Tchinda, N., Lv, X., Tasheh, S. N., Ghogomu, J. N., Du, L.: Atmospheric fate of organosulfates through gas-phase and aqueous-phase reactions with hydroxyl radicals: implications for inorganic sulfate formation. Atmos. Chem. Phys. 25, 8575-8590. 10.5194/acp-25-8575-2025, 2025.

Wu, Z., Wang, Y., Tan, T., Zhu, Y., Li, M., Shang, D., Wang, H., Lu, K., Guo, S., Zeng, L., Zhang, Y.: Aerosol Liquid Water Driven by Anthropogenic Inorganic Salts: Implying Its Key Role in Haze Formation over the North China Plain. Environmental Science & Technology Letters. 5, 160-166. 10.1021/acs.estlett.8b00021, 2018.

Zheng, B., Zhang, Q., Zhang, Y., He, K. B., Wang, K., Zheng, G. J., Duan, F. K., Ma, Y. L., Kimoto, T.: Heterogeneous chemistry: a mechanism missing in current models to explain secondary inorganic aerosol formation during the January 2013 haze episode in North China. Atmos. Chem. Phys. 15, 2031-2049. 10.5194/acp-15-2031-2015, 2015.

**Response to Referee#2**

Thanks to your careful reading and their constructive comments and suggestions on our manuscript. The reviewers' comments and suggestions are shown as *italicized font*, our response to the comments is normal font. New or modified text is in normal font and in blue. Details are as follows.

- - - - - - - - - - - - - - - - - - - - - - - - - - - - - - - - - - - - - - - - - - - - - - - - - - - - -

Referee's comments:

*This manuscript presents a valuable study on organosulfates (OSs) in PM2.5, combining molecular-level characterization, source apportionment, and OS–precursor correlation to improve OS classification in urban aerosols. The results reveal previously underestimated contributions of aliphatic OSs and nitrooxy OSs (NOSs) and highlight key formation factors such as aerosol liquid water content, inorganic sulfate, Ox, and pH. The study is scientifically sound and provides important insights into urban SOA formation. However, some methodological details (e.g., filter baking, extraction procedure, instrument calibration, and polyisotopic ion identification) require clarification to ensure reproducibility. After addressing these points, the manuscript would make a significant contribution to the field.*

[response]

Thanks for your comments. Please check our point-by-point response.

*Specific Comments*

*(1) Line 98 – The filters were generally baked at 550 ℃ before collection. Did the authors perform this step? This should be clarified.*

[response]

Thank you for your comment. All filters were pre-baked at 550 ℃ for 9 hours before sampling to remove the background organic matters.

[revised]

Lines 113–115:

The samples were stored in a freezer at -18 °C immediately after collection. The maximum duration between the completion of sampling and the start of chemical analysis was approximately 40 days.

*(2) Lines 112–114 – How many extraction cycles were conducted, and what volume of methanol was used? What were the extraction efficiencies of OSs from the filters? More detailed information about the extraction procedure would be helpful since OSs were quantitatively analyzed.*

[response]

Thank you for your thoughtful comment regarding the extraction procedure. In this study, each quartz fiber filter was extracted ultrasonically twice, with each extraction lasting 20 minutes. A total volume of 10 mL of LC-MS grade methanol was used per sample.

As for the extraction efficiencies of OSs, due to the fundamental characteristics of our non-target analysis (NTA) workflow, determining extraction efficiencies for individual OS presents a significant challenge. In the NTA approach, the molecular composition and concentration of OS are unknown prior to HPLC-HRMS analysis and subsequent data processing. Quantitative estimation relies on surrogate standards (Ma et al., 2022; Qiu et al., 2024), as authentic standards for the vast majority of detected OSs are unavailable. While we used specific surrogates for quantification of different OSs classifications, obtaining extraction efficiencies would require separate experiments with each corresponding authentic compound, which is currently impractical given the structural diversity encountered.

This inherent limitation of semi-quantification in NTA is acknowledged in the field and contributes to the overall analytical uncertainty (Nozière et al., 2015). To address this limitation, we have added relevant discussion in the revised manuscript. Our study's comparative conclusions on OS abundance and formation drivers across cities remain robust, as all samples were processed using the identical, rigorously controlled extraction and analysis approach.

[revised]

Line 120–124:

Sample extraction followed established protocols (Wang et al., 2020). Briefly, filters were ultrasonically extracted twice for 20 minutes. A total volume of 10 mL of LC-MS grade methanol (Merck Inc.) was used for each sample. All extracts were filtered through 0.22 μm PTFE syringe filters, and evaporated under a gentle stream of high-purity $N_2$ (>99.99%). The dried extracts were then redissolved in 2 mL of LC-MS grade methanol for analysis.

Line 413–421:

For NTA, the use of surrogate standards for quantification OS mass concentration introduced uncertainty, particularly due to the extraction efficiency of individual OSs species from quartz fiber filters could not be determined. Although we have adopted standardized extraction protocol ensures high comparability across our samples, absolute extraction recoveries may vary. In addition, this approach depends on public molecular composition such as mzCloud and ChemSpider integrated within the Compound Discoverer software, which contain limited entries for organosulfates. Reliance on these databases for compound identification may therefore underestimate OS mass

concentrations in urban environments. For example, OSs identified here accounted for less than 1% of total OA mass, whereas recent work (Ma et al., 2025) reported approximately 20% contributions.

*(3) Line 115 – Did the authors calibrate the instruments used in this study?*

[response]

Thanks for your comment. All instruments used in this study were calibrated to ensure the reliability of the measurement data. Specifically, the online gas and particulate monitors were calibrated with certified standard gases before and after the observation period, and underwent automatic zero/span checks every 24 hours (at 00:00 local time). Water-soluble ions (analyzed by MARGA-IC), OC/EC, and XRF systems were calibrated on a weekly basis using corresponding standard solutions or reference materials. The online GC-MS system used for VOC measurements was automatically calibrated every 24 hours using a multipoint standard gas mixture. These calibration procedures were consistently applied throughout the sampling campaign to maintain data quality and comparability. We have added a summary of these calibration protocols in the revised manuscript.

[revised]
Lines 134–139:

All instruments were calibrated to ensure the reliability of the measurement data. Specifically, the online gas pollutants and particulate matter automatic analyzers underwent automatic zero/span checks every 24 hours at 0:00 local time. For MARGA-ion chromatography, OC/EC analyzers, and XRF systems were calibrated weekly. The online GC-MS system was automatically calibrated every 24 hours using standard VOCs mixture.

*(4) Line 135 – Polyisotopic ions must be identified during molecular formula assignment to avoid misinterpretation of molecular compositions. It appears that the authors may not have performed this step.*

[response]

Thank you for your comment. We confirm that isotopic patterns, including polyisotopic ions, were indeed considered during the formula assignment process in the NTA workflow using Compound Discoverer software. Specifically, the algorithm accounts for the presence of $^{13}C$ isotope when matching the HRMS data. In line with common parameter setup in previous HRMS analysis for atmospheric organic aerosols (Ma et al., 2022; Qiu et al., 2024; Wang et al., 2017; Wang et al., 2019), up to one $^{13}C$ isotope was allowed in each molecule. This step helps avoid misassignment of identified compounds. We have revised relevant description in the revised manuscript.

[revised]
Lines 150–152:

Molecular formulas were assigned based on elemental combinations $C_cH_hO_oN_nS_s$ (c = 1–90, h = 1–200, o = 0–20, n = 0–1, s = 0–1) within a mass tolerance of 0.005 Da with up to one $^{13}C$ isotope.

*(5) Lines 194–200 – Were there any sulfur-containing compounds that could not be assigned? How did the authors calculate the total mass concentration of OSs?*

[response]

Thanks for your comment. Regarding the first part of your question, it is indeed possible that some sulfur-containing organic compounds were not classified as OSs in our analysis. As described in the manuscript (Section 2.2), our identification of OSs was based on two criteria: (1) a molecular formula containing sulfur, and (2) the presence of characteristic OS fragments in the corresponding $MS^2$ spectrum, specifically the hydrogensulfate ion ($HSO_4^-$, *m/z* 96.96010). Therefore, although several CHOS or CHONS compounds were detected, only those exhibiting these diagnostic fragments in their $MS^2$ spectra were definitively assigned as OSs. Some sulfur-containing organic molecules that did not show these fragments and were therefore not categorized as OSs. Furthermore, it is possible that some low-abundance OSs were not identified if their signal intensity was too low to trigger a reliable $MS^2$ acquisition during data-dependent analysis, which could lead to an underestimation. We acknowledge this as a common limitation in NTA.

Concerning the calculation of the total OS mass concentration, the value reported represents the sum of the (semi-)quantified concentrations for all OS species that met our identification criteria (see Section 2.3). In brief, different categories of OSs were quantified or semi-quantified using specific representative standards. The concentration of each detected OS was estimated based on its peak area relative to the calibration curve of its corresponding standard, and these individual concentrations were then summed to obtain the total OS mass concentration. We have revised this explanation in our revised manuscript.

[revised]
Lines 166–169:

It should be noted that several CHOS (composed of C, H, O, and S atoms, hereinafter) and CHONS species were not determined as OSs due to their low-abundance and insufficient to trigger reliable data-dependent $MS^2$ acquisition, which may lead to an underestimation of total OS mass concentration.

The total mass concentration of OS reported in this study is the sum of the (semi-)quantified concentrations of all individual OS species that met the identification criteria described in Section 2.3.

*(6) Line 209 – ALWC indeed plays an important role in OS formation. However, ALWC is also influenced by temperature and aerosol hygroscopicity in addition to relative humidity. Could the authors provide the specific ALWC values here? This would help readers better follow the discussion.*

[response]

Thanks for your comments. We have added the specific value of ALWC.

[revised]
Lines 268–270:

Considering persistently high RH (consistently >70%) during this period, as displayed in Figure S5, ALWC (117.9 μg/m$^3$ in average) therefore increased and facilitated the heterogeneous oxidation of $SO_2$ to particulate sulfate (Wang et al., 2016b; Ye et al., 2023).

*(7) Lines 213–215 – How did the authors determine this?*

[response]

Thanks for your comment. We identified the period of intense fireworks activity primarily through significant increases in the concentrations of recognized fireworks tracers, especially Ba and K, as displayed in Figure S4 in Supplementary Information. This surge in fireworks emissions led to a substantial emission of $SO_2$. Concurrently, high RH during this period led to elevated ALWC. The combination of high $SO_2$ levels and high ALWC is known to promote the heterogeneous oxidation of $SO_2$ to particulate sulfate (Wang et al., 2016; Ye et al., 2023), a process supported by the observed increase in sulfate concentrations during this period (see Figure S5 in Supplementary Information). Since particulate sulfate is a key reactant in OS formation pathways, its enhanced production provided a direct precursor for OS formation (Xu et al., 2024; Wang et al., 2020).

Furthermore, the fireworks activity led to increased concentrations of transition metals such as Fe and Mn (see Figure S4 in Supplementary Information). Based on established literature demonstrating the catalytic role of these metals in aqueous-phase radical chemistry and OS formation (Huang et al., 2019; Huang et al., 2018), we inferred that their increased presence likely provided an additional catalytic pathway, thereby further facilitating OS production during this episode.

We acknowledge that the role of transition metal catalysis, while strongly suggested by the correlated data and literature, is interpretive in the context of our ambient study. We have added relevant discussion in the revised manuscript.

[revised]
Lines 263–277:

This episode coincided with a period of intense fireworks activity, as evidenced by significant increases in the concentrations of recognized fireworks tracers, especially Ba and K (see Figure S4), leading to an increase in $SO_2$ emission. We noted that though K may originate from biomass burning, its trend in concentration shows good consistency with that of Ba. Therefore, we still infer that fireworks activity are also the primary source of K. Considering persistently high RH (consistently >70%) during this period, as displayed in Figure S5, ALWC (117.9 μg/m$^3$ in average) therefore increased and facilitated the heterogeneous oxidation of $SO_2$ to particulate sulfate (Wang et al., 2016b; Ye et al., 2023). Since particulate sulfate serves as a key reactant in OS formation pathways, its elevated concentration directly promoted OS production (Xu et al., 2024; Wang et al., 2020). Furthermore, fireworks activity led to concurrent increases in the concentrations of transition metals, notably Fe and Mn (Figure S4), which are known to catalyze aqueous-phase radical chemistry and OS formation (Huang et al., 2019; Huang et al., 2018a). Therefore, the pronounced OS mass concentration during this period is attributed to a combination of elevated precursor emissions ($SO_2$), high-RH conditions favoring aqueous-phase processing, and the potential catalytic role of co-emitted transition metals.

*(8) Lines 218–219 – The authors emphasize a new classification approach here and in the Introduction. However, this approach does not appear to show clear distinctions compared with previous methods. More detailed description of the current approach should be provided in the revised manuscript.*

[response]
Thanks for your comment. The "traditional classification methods" referred to in our study mainly denote the widely used framework in the literature for the identification and classification of OS, which is based on laboratory experiments and MS characteristics.

(1) **Precursor-controlled laboratory chamber and/or flow tube experiments.** In laboratory chamber or flow tube experiments, only specified target VOCs, such as isoprene, α-pinene, β-caryophyllene, toluene, and etc. are introduced, together with seed particles like ammonium sulfate under the oxidation of $O_3$ and OH radicals, so that the OS formed in the resulting SOA can be identified in a relatively simple system. For example, Surratt et al. used biogenic SOA smog chamber experiments to provide the

first clear evidence that the isoprene oxidation product IEPOX can further react with acidic sulfate particles to form typical isoprene-derived OSs such as $C_5H_{12}O_7S$ (Froyd et al., 2010; Surratt et al., 2007; Martinsson et al., 2017). Similar approach has also been applied to α-pinene and other monoterpene as well as sesquiterpene precursors, leading to the identification of a series of $C_{10}$ and $C_{15}$ OSs/NOSs species (Surratt et al., 2007; Hatch et al., 2011). From these experiments, several precursor-specific OSs are determined.

(2) **Establishing "tracer OS–precursor VOC" lists.** On the basis of the laboratory experiments and OS-precursor relationship proved by previous studies, a number of reviews and regional studies in recent years have systematically compiled those "tracer-type OS" that have been repeatedly confirmed, thereby forming OS lists categorized by precursor VOCs type. Table S4 in the Supplementary Information summarizes the OS-precursor relationship. For instance, $C_5H_{12}O_7S$ and $C_5H_8O_7S$ are widely accepted as representative OS formed via the high- and low-NOx pathways of isoprene/IEPOX, respectively; a suite of $C_{10}H_{18}O_5S$ and $C_{10}H_{16}NO_7S$ species are classified as monoterpene-derived OS/NOS; and some $C_2$–$C_3$ small-molecule OS (e.g., glycolic acid sulfate) are generally regarded as arising from mixed contributions of small carbonyls and carboxylic acids (Brüggemann et al., 2020; Fan et al., 2022; Tsona Tchinda et al., 2025).

However, we found significant limitation when we used this approach to classify identified OSs, as the "precursor-specific OS" relationship was not established for numerous of OSs. Therefore, a large number of OSs was classified into "Unknown OSs". In addition, this approach is relying solely on a small set of parameters such as DBE and H/C typically only allows a coarse separation between broad aliphatic and aromatic classes, making it difficult to achieve a more refined source classification for structurally complex OS. In order to classify those "Unknown OSs", we used an approach from multiple perspectives, including molecular-composition metrics, precursor based-PMF source apportionment, and correlation analysis between OS and their putative precursors.

In summary, compared with traditional "precursor-specific OS" classification approach, our classification approach realized accurate classification of detected OSs.

[revised]

Lines 178–189:

To classify the identified OSs, we employed and compared two distinct classification approaches. Firstly, a conventional classification approach relies primarily on precursor–product

relationships established through controlled laboratory chamber experiments and field campaigns (Zhao et al., 2018; Wang et al., 2021; Deng et al., 2021; Xu et al., 2021; Mutzel et al., 2015; Brüggemann et al., 2020; Yang et al., 2024; Duporté et al., 2020; Huang et al., 2023b; Wang et al., 2022b; Riva et al., 2016a). Based on these established precursor–product relationships, detected OSs and NOSs were classified into four groups: Monoterpene OSs (including Monoterpene NOSs, hereinafter), Aliphatic OSs (including Aliphatic NOSs, hereinafter), Aromatic OSs (including Aromatic NOSs, hereinafter), and Sesquiterpene OSs (including Sesquiterpene NOSs, hereinafter) (see Table S4 for details). It is apparently that this approach has notable limitations when applied to detected OS in atmospheric aerosols. A substantial fraction of detected OSs does not match known laboratory tracers and are thus labeled Unknown OSs (including Unknown NOSs, hereinafter).

*(9) Line 221 – Each numerical value shown here should correspond to the respective city.*

[response]

    Thanks for your suggestion. We have revised accordingly.

[revised]
Lines 288–291:

    As displayed in Figure 2(b), Monoterpene OSs dominated detected OSs across all cities, contributing 55.2% (Beijing), 46.8% (Taiyuan), and 72.3% (Changsha) to total OS, respectively. Biogenic-emitted monoterpene is the precursor of Monoterpene OSs.

**References**

Brüggemann, M., Xu, R., Tilgner, A., Kwong, K. C., Mutzel, A., Poon, H. Y., Otto, T., Schaefer, T., Poulain, L., Chan, M. N., Herrmann, H.: Organosulfates in Ambient Aerosol: State of Knowledge and Future Research Directions on Formation, Abundance, Fate, and Importance. Environmental Science & Technology. 54, 3767-3782. 10.1021/acs.est.9b06751, 2020.

Eddingsaas, N. C., VanderVelde, D. G., Wennberg, P. O.: Kinetics and Products of the Acid-Catalyzed Ring-Opening of Atmospherically Relevant Butyl Epoxy Alcohols. The Journal of Physical Chemistry A. 114, 8106-8113. 10.1021/jp103907c, 2010.

Fan, W., Chen, T., Zhu, Z., Zhang, H., Qiu, Y., Yin, D.: A review of secondary organic aerosols formation focusing on organosulfates and organic nitrates. Journal of Hazardous Materials. 430, 128406. https://doi.org/10.1016/j.jhazmat.2022.128406, 2022.

Froyd, K. D., Murphy, S. M., Murphy, D. M., de Gouw, J. A., Eddingsaas, N. C., Wennberg, P. O.: Contribution of isoprene-derived organosulfates to free tropospheric aerosol mass. Proceedings of the National Academy of Sciences. 107, 21360-21365. 10.1073/pnas.1012561107, 2010.

Hatch, L. E., Creamean, J. M., Ault, A. P., Surratt, J. D., Chan, M. N., Seinfeld, J. H., Edgerton, E. S., Su, Y., Prather, K. A.: Measurements of Isoprene-Derived Organosulfates in Ambient Aerosols by Aerosol Time-of-Flight Mass Spectrometry - Part 1: Single Particle Atmospheric Observations in Atlanta. Environmental Science & Technology. 45, 5105-5111. 10.1021/es103944a, 2011.

Huang, L., Cochran, R. E., Coddens, E. M., Grassian, V. H.: Formation of Organosulfur Compounds through Transition Metal Ion-Catalyzed Aqueous Phase Reactions. Environmental Science & Technology Letters. 5, 315-321. 10.1021/acs.estlett.8b00225, 2018.

Huang, L., Coddens, E. M., Grassian, V. H.: Formation of Organosulfur Compounds from Aqueous Phase Reactions of S(IV) with Methacrolein and Methyl Vinyl Ketone in the Presence of Transition Metal Ions. ACS Earth and Space Chemistry. 3, 1749-1755. 10.1021/acsearthspacechem.9b00173, 2019.

Iinuma, Y., Müller, C., Berndt, T., Böge, O., Claeys, M., Herrmann, H.: Evidence for the Existence of Organosulfates from β-Pinene Ozonolysis in Ambient Secondary Organic Aerosol. Environmental Science & Technology. 41, 6678-6683. 10.1021/es070938t, 2007.

Ma, J., Ungeheuer, F., Zheng, F., Du, W., Wang, Y., Cai, J., Zhou, Y., Yan, C., Liu, Y., Kulmala, M., Daellenbach, K. R., Vogel, A. L.: Nontarget Screening Exhibits a Seasonal Cycle of PM2.5 Organic Aerosol Composition in Beijing. Environmental Science & Technology. 56, 7017-7028. 10.1021/acs.est.1c06905, 2022.

Martinsson, J., Monteil, G., Sporre, M. K., Kaldal Hansen, A. M., Kristensson, A., Eriksson Stenström, K., Swietlicki, E., Glasius, M.: Exploring sources of biogenic secondary organic aerosol compounds using chemical analysis and the FLEXPART model. Atmos. Chem. Phys. 17, 11025-11040. 10.5194/acp-17-11025-2017, 2017.

Nozière, B., Kalberer, M., Claeys, M., Allan, J., D'Anna, B., Decesari, S., Finessi, E., Glasius, M., Grgić, I., Hamilton, J. F., Hoffmann, T., Iinuma, Y., Jaoui, M., Kahnt, A., Kampf, C. J., Kourtchev, I., Maenhaut, W., Marsden, N., Saarikoski, S., Schnelle-Kreis, J., Surratt, J. D., Szidat, S., Szmigielski, R., Wisthaler, A.: The Molecular Identification of Organic Compounds in the Atmosphere: State of the Art and Challenges. Chemical Reviews. 115, 3919-3983. 10.1021/cr5003485, 2015.

Qiu, Y., Qiu, T., Wu, Z., Liu, Y., Fang, W., Man, R., Liu, Y., Wang, J., Meng, X., Chen, J., Liang, D., Guo, S., Hu, M.: Observational Evidence of Brown Carbon Photobleaching in Urban Atmosphere at Molecular Level. Environmental Science and Technology Letters. 11, 1032-1039. 10.1021/acs.estlett.4c00647, 2024.

Riva, M., Da Silva Barbosa, T., Lin, Y. H., Stone, E. A., Gold, A., Surratt, J. D.: Chemical characterization of organosulfates in secondary organic aerosol derived from the photooxidation of alkanes. Atmospheric Chemistry and Physics. 16, 11001-11018. 10.5194/acp-16-11001-2016, 2016.

Surratt, J. D., Kroll, J. H., Kleindienst, T. E., Edney, E. O., Claeys, M., Sorooshian, A., Ng, N. L., Offenberg, J. H., Lewandowski, M., Jaoui, M., Flagan, R. C., Seinfeld, J. H.: Evidence for Organosulfates in Secondary Organic Aerosol. Environmental Science & Technology. 41, 517-527. 10.1021/es062081q, 2007.

Tsona Tchinda, N., Lv, X., Tasheh, S. N., Ghogomu, J. N., Du, L.: Atmospheric fate of organosulfates through gas-phase and aqueous-phase reactions with hydroxyl radicals: implications for inorganic sulfate formation. Atmos. Chem. Phys. 25, 8575-8590. 10.5194/acp-25-8575-2025, 2025.

Wang, G., Zhang, R., Gomez, M. E., Yang, L., Levy Zamora, M., Hu, M., Lin, Y., Peng, J., Guo, S., Meng, J., Li, J., Cheng, C., Hu, T., Ren, Y., Wang, Y., Gao, J., Cao, J., An, Z., Zhou, W., Li, G., Wang, J., Tian, P., Marrero-Ortiz, W., Secrest, J., Du, Z., Zheng, J., Shang, D., Zeng, L., Shao, M., Wang, W., Huang, Y., Wang, Y., Zhu, Y., Li, Y., Hu, J., Pan, B., Cai, L., Cheng, Y., Ji, Y., Zhang, F., Rosenfeld, D., Liss, P. S., Duce, R. A., Kolb, C. E., Molina, M. J.: Persistent sulfate formation from London Fog to Chinese haze. Proceedings of the National Academy of Sciences. 113, 13630-13635. 10.1073/pnas.1616540113, 2016.

Wang, Y., Hu, M., Lin, P., Guo, Q., Wu, Z., Li, M., Zeng, L., Song, Y., Zeng, L., Wu, Y., Guo, S., Huang, X., He, L.: Molecular Characterization of Nitrogen-Containing Organic Compounds in Humic-like Substances Emitted from Straw Residue Burning. Environmental Science & Technology. 51, 5951-5961. 10.1021/acs.est.7b00248, 2017.

Wang, Y., Hu, M., Lin, P., Tan, T., Li, M., Xu, N., Zheng, J., Du, Z., Qin, Y., Wu, Y., Lu, S., Song, Y., Wu, Z., Guo, S., Zeng, L., Huang, X., He, L.: Enhancement in Particulate Organic Nitrogen and Light Absorption of Humic-Like Substances over Tibetan Plateau Due to Long-Range Transported Biomass Burning Emissions. Environmental Science & Technology. 53, 14222-14232. 10.1021/acs.est.9b06152, 2019.

Wang, Y., Hu, M., Wang, Y.-C., Li, X., Fang, X., Tang, R., Lu, S., Wu, Y., Guo, S., Wu, Z., Hallquist, M., Yu, J. Z.: Comparative Study of Particulate Organosulfates in Contrasting Atmospheric Environments: Field Evidence for the Significant Influence of Anthropogenic Sulfate and NOx. Environmental Science & Technology Letters. 7, 787-794. 10.1021/acs.estlett.0c00550, 2020.

Xu, R., Chen, Y., Ng, S. I. M., Zhang, Z., Gold, A., Turpin, B. J., Ault, A. P., Surratt, J. D., Chan, M. N.: Formation of Inorganic Sulfate and Volatile Nonsulfated Products from Heterogeneous Hydroxyl Radical Oxidation of 2-Methyltetrol Sulfate Aerosols: Mechanisms and Atmospheric Implications. Environmental Science & Technology Letters. 11, 968-974. 10.1021/acs.estlett.4c00451, 2024.

Ye, C., Lu, K., Song, H., Mu, Y., Chen, J., Zhang, Y.: A critical review of sulfate aerosol formation mechanisms during winter polluted periods. Journal of Environmental Sciences. 123, 387-399. https://doi.org/10.1016/j.jes.2022.07.011, 2023.

**Response to Referee#3**

Thanks to your careful reading and their constructive comments and suggestions on our manuscript. The reviewers' comments and suggestions are shown as *italicized font*, our response to the comments is normal font. New or modified text is in normal font and in blue. Details are as follows.
* * *
Referee's comments:

*This study by Qiu et al. investigates organosulfates in aerosols in three Chinese megacities, Beijing, Taiyuan, and Changsha, using an integrated framework that combines LC-Orbitrap mass spectrometry with non-targeted analysis. Additionally, precursor-constrained PMF analysis, together with meteorological data (RH, ALWC, sulfate, Ox), were incorporated to provide a detailed interpretation of the formation pathways. It is interesting that the authors installed a denuder upstream of the sampler to remove SO₂ during sampling, which helps demonstrate that the detected organosulfates are not solely sampling artifacts, a concern raised in previous studies. The novel integration of nontarget analysis of HPLC-Orbitrap mass spectrometry data with precursor-constrained PMF offers an alternative and informative approach for classifying organosulfates from a molecular perspective, representing a valuable methodological contribution to the field. Overall, the manuscript is well written and well structured, however there are several grammatical errors which should be corrected. The comments below should be carefully addressed before the manuscript can be fully considered for publication in Atmospheric Chemistry and Physics.*

[response]

      Thanks for your comments. Please check our point-by-point response.

*General comments*

*(1) Sampling was conducted only from December 2023 to January 2024. However, organosulfate formation is highly seasonal, particularly for pathways driven by biogenic emissions and photochemistry. The authors should discuss that the results may not represent annual OS behaviour.*

[response]

      We agree that the formation pathways and abundances of OSs exhibit significant seasonal variations. Our field campaign was conducted in winter to investigate OS formation under conditions typically characterized by elevated anthropogenic emissions, lower temperatures, and specific meteorological patterns such as frequent haze episodes with high RH in urban regions. Therefore, the main conclusion in this

study, including the underestimation of Aliphatic OSs and NOSs, and their formation driving factors are contextualized within the winter season and may not be directly extrapolated to represent annual OS behavior.

To address your concern and ensure clarity for readers, we have added relevant discussion in the "Conclusions and Implications" section of the revised manuscript.

[revised]

Lines 405–412:

However, this study still faces several challenges. This work was conducted during the winter. OS formation exhibits seasonal variability, particularly for pathways driven by biogenic VOCs emissions and photochemical activity, which are generally enhanced in warmer months. Hence, the underestimation of Aliphatic OSs, and their key formation factors determined in this work remain valid insights for the winter period but may not fully represent annual OS behavior. Future long-term observations are necessary to resolve the complete annual cycle of OS composition, quantify the shifting contributions of anthropogenic versus biogenic precursors, and understanding how key formation driving factors evolve with changing atmospheric conditions.

*(2) The precursor-constrained PMF uses only a limited set of VOCs. which is reasonable. However, many OSs come from multiple or unknown precursors, so the source assignments may not be very stable. The authors should discuss how VOC correlations might affect the results and the possibility of misclassification.*

[response]

We agree that many OSs come from multiple or unknown VOCs precursors. we acknowledge that this limited set cannot represent the full spectrum of potential VOCs precursors. Therefore, in order to avoid the possibility of misclassification, we did not rely on the VOCs correlations. Our methodology integrated following lines of evidence to minimize misclassification:

(1) **Multiple precursor-constrained PMF.** In addition to tracer VOCs, we incorporated auxiliary OS tracers with well-established precursor into the PMF model. These OSs were selected due to their high detection frequency and unambiguous source linkage ($C_{11}H_{22}O_5S$ and $C_{12}H_{24}O_5S$ for Aliphatic OSs/NOSs; $C_{10}H_{10}O_7S$ and $C_{11}H_{14}O_7S$ for Aromatic OSs/NOSs; $C_{10}H_{18}O_5S$, $C_{10}H_{17}NO_7S$ for Monoterpene OSs/NOSs; $C_{14}H_{28}O_6S$, $C_{15}H_{25}NO_7S$ for Sesquiterpene OSs/NOSs). These OSs showed similar source profiles with corresponding tracer VOCs, thereby strengthening the source specificity and stability of the resolved profiles.

(2) **Validation via diagnostic MS$^2$ fragmentation patterns.** For OSs classified into different categories, we examined their MS$^2$ spectra. Those OSs in a certain category showed similar fragmentation patterns in their MS$^2$ spectra. Aliphatic OSs and

NOSs showed sequential alkyl chain cleavages ($\Delta$ $m/z$ = 14.0157) and saturated alkyl fragments ($[C_nH_{2n+1}]^-$ or $[C_nH_{2n-1}]^-$); Monoterpene OSs and NOSs displayed $[C_nH_{2n-3}]^-$ fragments; Aromatic OSs/NOSs exhibited characteristic aromatic substituent fragments ($[C_6H_5R-H]^-$, R = alkyl, carbonyl, -OH, or H). These fragment patterns confirm the reliability of this classification approach.

We acknowledge that absolute certainty for every individual OS is unattainable in such a complex system, and some uncertainty remains. However, the convergence of evidence from precursor-constrained PMF and structural $MS^2$ validation reduces the likelihood of systematic misclassification.

[revised]

Lines 241–244:

While absolute certainty for every individual OS in a complex ambient mixture is unattainable, integrating the precursor-constrained PMF model, tracer VOCs correlation analysis, and $MS^2$ fragment patterns validation significantly reduces the likelihood of systematic misclassification.

*(3) Discussions of formation pathways mainly rely on correlations with RH, ALWC, sulfate, and Ox. While these correlations are informative, they do not establish mechanistic causality. Strong statements about OSs formation (e.g., heterogeneous formation, as noted throughout the manuscript) would be better presented as hypotheses supported by observational evidence, rather than as definitive mechanisms.*

[response]

We acknowledge that the discussion regarding the Ox on Aliphatic OSs and NOSs formation is inferential. However, these discussion is based on mechanistic insights from previous laboratory studies. The offline measurement techniques to directly identify these chemically unstable, short-lived intermediates such as epoxides or Criegee intermediates in ambient aerosol samples is indeed challenging or nearly impossible. The aim of this paragraph is to explore how atmospheric oxidation capacity (represented by Ox concentration) impact the formation of Aliphatic OSS/NOSs, and logically connect field observational evidence to the laboratory-supported chemical mechanisms.

From a mechanistic perspective, laboratory studies have clearly demonstrated that enhanced atmospheric oxidation capacity promotes the formation of cyclic intermediates, and that subsequent acid-catalyzed ring-opening is a crucial pathway for OS formation (Eddingsaas et al., 2010; Iinuma et al., 2007; Riva et al., 2016). Our observational results correlations are consistent with and supportive of such a pathway operating in the urban atmosphere. In the revised manuscript, we have added more

discussion in our revised manuscript.

[revised]
Lines 385–391:

Previous laboratory studies have suggested that enhanced atmospheric oxidation capacities promote the oxidation of VOCs (Zhang et al., 2022; Wei et al., 2024), forming cyclic intermediates. We therefore inferred that the increase in Ox facilitates the formation of cyclic intermediates derived from long-chain alkenes. Subsequent acid-catalyzed and ring-opening reactions are important pathways of heterogeneous OSs formation, including Aliphatic OSs (Eddingsaas et al., 2010; Iinuma et al., 2007; Brüggemann et al., 2020).

*(4) This work quantifies organosulfates using a few surrogate standards. Table 1 lists four standards, but it is stated that only compounds with more than 8 carbon atoms are quantified, thus one standard is excluded. However, as reported by e.g. Ma et al. 2025, Nat. Commun., the structural diversity of organosulfates leads to varying ionization efficiencies, introducing significant uncertainty in absolute concentrations. A more detailed discussion of these uncertainties is needed to strengthen the quantitative interpretation.*

[response]

We agree with your comments in noting a point that requires clarification: while our quantitative analysis focused on OSs with C ≥ 8, the standard phenyl sulfate ($C_6H_5O_4S^-$) was used as a surrogate for Aromatic OSs/NOSs and for unknown OSs/NOSs eluting with the RT 0.50–2.00 min. This choice was based on its structural analogy to Aromatic OSs and its stable, well-defined chromatographic peak, making it the most suitable available surrogate for that compound class despite the carbon number discrepancy.

We fully acknowledge that using surrogate standards introduces significant uncertainty into absolute concentration estimates, a well-known challenge in NTA of complex mixtures like OSs. As highlighted by Ma et al. (2025) and others, differences in molecular structure, functional groups, and backbone chemistry between a surrogate and the target analytes do lead to substantial variations in ESI efficiency, which is a major source of quantitative error in LC-MS-based methods. We have significantly expanded the discussion of these quantitative uncertainties in the revised manuscript.

[revised]
Lines 200–205:

This quantification approach introduces inherent uncertainty, as differences in molecular

structure and functional groups between a surrogate and detected OSs have different ionization efficiency (Ma et al., 2025), which is a well-documented challenge in NTA of complex mixtures. However, this approach provides a consistent basis for comparing the relative abundance of OS in different cities and their formation driving factors. Hence, the mass concentration of detected OSs is still reliable in understanding their classification and formation driving factors.

*(5) More details should be included in the experimental section. This includes e.g. aerosol collection (type, flow rate) and the denuder (including efficiency in removal of SO2).*

[response]

Thanks for your suggestion. We have expanded Section 2.1 to provide a more comprehensive description of the aerosol collection and pretreatment procedures, including the pre-treatment of filters, sampling type, flow rate, and etc. However, the SO$_2$ removal efficiency cannot be provided, as we used a commercial denuder coated with NaCl/Na$_2$CO$_3$ to remove SO$_2$. While we did not perform an independent, in-field calibration of its SO$_2$ removal efficiency for this campaign, the design and coating are well-established for effective acidic gas removal.

[revised]
Lines 105–111:

Daily PM$_{2.5}$ samples were collected on quartz fiber filters ($\varphi$ = 47 mm, Whatman Inc.) from 9:00 to 8:00 local time the next day. All quartz fiber filters were pre-baked at 550 °C before sampling to remove the background organic matters. In Beijing and Taiyuan, RH-resolved sampling was performed using an RH-resolved sampler, stratifying daily samples into low (RH ≤ 40%), moderate (40% < RH ≤ 60%), and high (RH > 60%) RH regimes with the sampling flow rate of 38 L/min. Due to persistently high RH in Changsha, a four-channel sampler (TH-16, Wuhan Tianhong Inc.) collected PM$_{2.5}$ samples without RH stratification with the flow rate of 16.7 L/min.

*Specific Comments:*

*(1) In the title "Revealing the" could be removed.*

[response]
Removed.

*(2) L22: "Organosulfates (OSs) are important component of" -> components. You define OSs as the plural of organosulfates, however in many instances it is better to use the singular e.g. line 27 OS precursor. This is often the case when OS is part of a composed wording (such as this example).*

[response]

Thank you for the valid grammatical suggestion regarding the use of "OS" and "OSs" in our manuscript. We fully agree with your point. In the revised version, we have standardized the usage throughout the text: the singular form "OS" is used when it functions as a modifier in compound terms, whereas the plural "OSs" is reserved when referring explicitly to multiple, discrete OSs compounds or species.

*(3) L24. It is not clear to the reader what you mean by "classification" here.*

[response]

Thanks for your comment. In our manuscript, "classification" specifically refers to the process of assigning the detected OSs to specific source categories based on their VOC precursors. This is achieved by integrating their molecular composition data with precursor-constrained PMF source apportionment and correlation analysis. We have revised our manuscript to clarify this definition, particularly in the abstract and introduction.

[revised]
Lines 23–29:

However, molecular composition, precursor-OS correspondence, and formation driving factors of OSs at different atmospheric conditions have not been fully constrained. In this work, we integrated OS molecular composition, precursor-constrained positive matrix factorization (PMF) source apportionment, and OS-precursor correlation analysis to classify OS detected from $PM_{2.5}$ samples according to their volatile organic compounds (VOCs) precursors collected from three different cities (Beijing, Taiyuan, and Changsha) in China.

*(4) L28. Please add "China" after the list of cities.*

[response]

Revised.

*(5) L39. What do you mean by "indicative role"?*

[response]

Thank you for pointing out the unclear terminology. We agree that the phrase "indicative role" was inappropriate. In the revised manuscript, we have rephrased the concluding statement in the Abstract to more directly and accurately reflect our findings.

[revised]
Lines 39–41:

These results reveal a significant underestimation of OSs derived from anthropogenic

emissions, particularly Aliphatic OSs, highlighting the need for a deeper understanding of SOA formation and composition in urban environments.

*(6) L55: vital seems like a strange word here.*

[response]

  Revised.

*(7) L64. This seems like a very strong statement given the uncertainties of the current approach.*

[response]

  Revised

[revised]
Lines 64–65:

  Classifying OS based on their precursors is a powerful approach for understanding OS formation from a mechanistic perspective.

*(8) L75-76: "For instance, increased aerosol liquid water content (ALWC) enhances OSs formation by promoting the uptake of gaseous precursors (Edwards et al., 2017; Brown et al., 2012)". Both papers report on NOx so it is not clear how they support this statement.*

[response]

  We are sorry for the wrong citation. These citations have been replaced.

*(9) L82 NTA is already defined.*

[response]

  Revised

*(10) L84. State that the cities are in China.*

[response]

  Revised

*(11) L93: Could the authors provide more information on why they conducted their research in these three cities? Can the authors discuss the extent to which organosulfate formation mechanisms can (or cannot) be generalized across different locations?*

[response]

  The selection of the three sampling cities, i.e., Beijing, Taiyuan, and Changsha, was deliberate, based on their representative differences in meteorological conditions

and primary emission profiles during winter.

From a meteorological perspective, Beijing and Taiyuan in northern China experience cold, dry winters with low RH. In contrast, Changsha, locates in the humid subtropical region of southern China, is characterized by relatively higher winter RH. This pronounced north-south humidity gradient allows us to explicitly investigate the role of ALWC, which is a critical factor in aqueous-phase OS formation (Cheng et al., 2016; Wu et al., 2018; Zheng et al., 2015).

Regarding the sources of $PM_{2.5}$, the three cities represent distinct dominant $PM_{2.5}$ source regimes. Coal combustion and industrial are two important $PM_{2.5}$ sources in Taiyuan, which is a traditional industrial and coal-mining base. Changsha's $PM_{2.5}$ profile is more dominated by vehicle emissions and domestic cooking. Beijing presents a complex mix where secondary formation processes become the predominant source of $PM_{2.5}$. Figure S4 in Supplementary Information shows the PMF results of $PM_{2.5}$ across three cities. These differences in primary emission types lead to varying abundances of key OS precursors, such as VOCs species and $SO_2$.

In summary, both meteorological conditions (particularly RH) and precursor emission profiles are major drivers of spatial heterogeneity in OS molecular composition and concentration. Therefore, selecting these three cities is suitable to investigate how these key variables govern OSs formation mechanisms across diverse urban environments in China.

[revised]
Lines 93–100:

The site selection was based on contrasts in winter meteorological conditions and dominate $PM_{2.5}$ sources. For meteorological conditions, Beijing and Taiyuan represent northern Chinese cities with cold, dry conditions (low RH). In comparison, Changsha is characterized by relatively higher winter RH. In terms of $PM_{2.5}$ sources, Taiyuan is a traditional industrial and coal-mining base, Changsha's pollution profile is more influenced by traffic and domestic cooking emissions, whereas Beijing is characterized by a high mass fraction of secondary aerosols. This enables a comparative analysis of OS formation mechanisms under varied atmospheric conditions.

*(12) L108: This was probably not 40 days for all filters? What is the reason for 24 h equilibration of filters, with potential for artefact formation?*

[response]
Thank you for your comments. Regarding the storage duration, "40 days" is an approximate estimate of the maximum time elapsed between the sampling during field campaign was completed and chemical analysis. It does not represent a uniform storage

period for every sample. All samples were immediately stored at -18 °C after collection and remained frozen until extraction. Concerning the 24-hour equilibration, this step is a standard procedure in our protocol to allow the frozen filters to acclimatize to a stable, controlled temperature (20 ± 1 °C) and relative humidity (40-45%) environment within a clean bench before analysis. The purpose is to minimize condensation during handling, not to "recover" any atmospheric state. While we acknowledge the theoretical potential for artefact formation during the handling step, the conditions during equilibration (moderate RH, absence of reactive gases, and darkness) are not conducive to the major pathways for OS formation. Therefore, we consider its contribution to artefactual OS formation to be negligible.

[revised]
Lines 114–118:

The maximum duration between the completion of sampling and the start of chemical analysis was approximately 40 days. Prior to analysis, all samples were equilibrated for 24 hours under controlled temperature (20 ± 1 °C) and RH (40-45%) within a clean bench, in order to allow the filters to reach a stable, reproducible condition for subsequent handling and to minimize moisture condensation.

*(13) L112: Please make sure to be specific so you need to state that extracts (not filters) were filtered.*

[response]
Revised.

[revised]
Lines 122–123:

All extracts were filtered through 0.22 μm PTFE syringe filters, and evaporated under a gentle stream of high-purity $N_2$ (>99.99%).

*(14) L114: Why are the samples redissolved in pure methanol? This could be the cause of the issue with analysis of early-eluting compounds. Please comment on this.*

[response]
Thanks for your comment. In our study, the main reason for the drying and redisslove was to concentrate the extracts. The initial ultrasonic extraction required a relatively large volume of methanol (10 mL per sample) to ensure efficient contact and recovery. This dilute extract would have resulted in low sensitivity for trace-level OSs. Redisslove in a small volume (2 mL) of the same solvent was therefore a necessary step to achieve detectable concentrations for a OSs in low concentrations.

We acknowledge that pure methanol may be not an ideal solvent for the hydrophilic interaction that likely governs the retention of the most polar OSs on our C18 column. This is a limitation of reversed-phase chromatography for highly polar species, leading to challenges in precisely quantifying some early-eluting species like small OSs with C ≤ 7. However, for the mid- to late-eluting OSs which are the focus in this study (C ≥ 8), the use of a consistent redisslove solvent across all samples also allows the comparisons between samples.

[revised]
Lines 175–177:

Though redisslove using pure methanol may not be the ideal solvent for retaining polar, early-eluting compounds on the reversed-phase column, it provided a consistent solvent for the analysis of the mid- and non-polar OS species (C ≥ 8) that are the focus of this study.

*(15) L135-136: s = 0-1!*

[response]
    Revised.

*(16) L144-145: Organosulfates primarily fragment under MS2 to produce ions would be HSO4− and SO4•− whereas HSO3− and SO3•− are characteristic of organosulfonates.*

[response]
    We sincerely apologize for the error in our manuscript. We have re-examining all our raw $MS^2$ data in response to your comment, we confirm that our actual data processing and screening were correctly based on the presence of the $HSO_4^-$ fragment. Therefore, the identification, classification, and all quantitative results for the OSs reported in our study remain entirely valid and unchanged. The error was confined to the textual description.

[revised]
Lines 163–164:

OSs were selected based on compounds with O/S ≥ 4 and $HSO_4^-$ (*m/z* 96.96010) fragments were observed in their corresponding $MS^2$ spectra.

*(17) L147: Although nitrooxy-organosulfates contain nitrogen, they are still a subtype of organosulfates. Since the authors discuss them together throughout the manuscript, it may be clearer to group both nitrooxy-organosulfates and organosulfates under the general category of "organosulfates." This would also simplify the presentation of*

*Figure 2.*

[response]

Revised.

*(18) L149: While reversed-phase columns can struggle with highly polar, very small organosulfates, I do not agree that they inherently have limitations for species as large as C7. In practice, such limitations are typically restricted to OSs with very short carbon chains (e.g., C1–C3), where retention becomes insufficient. Moreover, the authors themselves used a C6 organosulfate as a standard, which further suggests that the method is capable of measuring species in this size range. Or is this issue due to the pure methanol solvent?*

[response]

Thanks for your comment. We focused the analysis on OSs with C ≥ 8 was driven less by a fundamental chromatographic inability to detect smaller species, and more by a practical assessment of data reliability of OSs detected by NTA in atmospheric PM$_{2.5}$. The core issue with C6-C7 OSs is not their detectability, but their highly ambiguous precursor-OS correspondence. In precursor-specific laboratory experiments, C6-C7 OSs have not been reported as major products. Therefore, their molecular formulas often lack unambiguous precursor assignments, making them difficult to classify reliably in our study. Furthermore, these more polar OSs generally have earlier RT, making their semi-quantification based on a single surrogate standard   particularly uncertain. The threshold of C ≥ 8 was chosen to ensure that the OSs included in our quantitative and classification. We have added more discussion in the revised manuscript.

[revised]
Lines 171–75:

To ensure the reliability of quantitative analysis and source attribution, this study focuses on OS species with C ≥ 8. The exclusion of smaller OSs (C ≤ 7) is based on challenges in their unambiguous identification, including co-elution with interfering compounds (Liu et al., 2024), and higher uncertainty in precursor assignment due to the lack of characteristic "tracer" molecules in laboratory experiments.

*(19) L178: In some instances it is 8 groups and in others 7. Please be consistent.*

[response]

Revised.

*(20) L183-184: This sentence needs clarification.*

[response]

    Revised.

    Correlation coefficients between classified OSs and corresponding VOCs (Monoterpene OSs vs. isoprene; Aromatic OSs vs. benzene; Aliphatic OSs vs. n-dodecane; Sesquiterpene OSs vs. isoprene) were calculated as a statistical auxiliary variable to verify the reliability of PMF results.

*(21) L196: Give the variation, one decimal is more appropriate here.*

[response]

    Revised.

*(22) L212-217: This section is quite speculative, which should be better reflected in the text. The time series of metals from fireworks show shorter spikes in concentrations, while sulfate and organosulfate concentrations are elevated for longer periods. K can also stem from biomass combustion.*

[response]

    Thanks for your comment. We agree that the original discussion could be perceived as overly definitive given the complexities in the data. We identified the period of intense fireworks activity primarily through significant increases in the concentrations of recognized fireworks tracers, especially Ba and K, as displayed in Figure S4 in the Supplementary Information. Though K may originate from biomass burning, its trend in concentration shows good consistency with that of Ba. Therefore, we still infer that fireworks activity are also the primary source of K. This surge in fireworks emissions led to a substantial emission of $SO_2$. Concurrently, high RH during this period led to elevated ALWC. The combination of high $SO_2$ levels and high ALWC is known to promote the heterogeneous oxidation of $SO_2$ to particulate sulfate (Wang et al., 2016; Ye et al., 2023), a process supported by the observed increase in sulfate concentrations during this period (Figure S5). Since particulate sulfate is a key reactant in OS formation pathways, its enhanced production provided a direct precursor for OS formation (Xu et al., 2024; Wang et al., 2020).

    Furthermore, the fireworks activity led to increased concentrations of transition metals such as Fe and Mn (see Figure S4). Based on established literature demonstrating the catalytic role of these metals in aqueous-phase radical chemistry and OS formation (Huang et al., 2019; Huang et al., 2018), we inferred that their increased presence likely provided an additional catalytic pathway, thereby further facilitating OS

**R12**

production during this episode.

[revised]

Lines 263–277:

This episode coincided with a period of intense fireworks activity, as evidenced by significant increases in the concentrations of recognized fireworks tracers, especially Ba and K (see Figure S4), leading to an increase in $SO_2$ emission. We noted that though K may originate from biomass burning, its trend in concentration shows good consistency with that of Ba. Therefore, we still infer that fireworks activity are also the primary source of K. Considering persistently high RH (consistently >70%) during this period, as displayed in Figure S5, ALWC therefore increased and facilitated the heterogeneous oxidation of $SO_2$ to particulate sulfate (Wang et al., 2016b; Ye et al., 2023). Since particulate sulfate serves as a key reactant in OS formation pathways, its elevated concentration directly promoted OS production (Xu et al., 2024; Wang et al., 2020). Furthermore, fireworks activity led to concurrent increases in the concentrations of transition metals, notably Fe and Mn (Figure S4), which are known to catalyze aqueous-phase radical chemistry and OS formation (Huang et al., 2019; Huang et al., 2018a). Therefore, the pronounced OS mass concentration during this period is attributed to a combination of elevated precursor emissions ($SO_2$), high-RH conditions favoring aqueous-phase processing, and the potential catalytic role of co-emitted transition metals.

*(23) L219: It is not clear what you mean by "conventional". Please be specific.*

[response]

Thanks for your comment. The "traditional classification methods" referred to in our study mainly denote the widely used framework in the literature for the identification and classification of OS, which is based on laboratory experiments and MS characteristics. We have clarified in our revised manuscript.

[revised]

Lines 178–189:

To classify the identified OSs, we employed and compared two distinct classification approaches. Firstly, a conventional classification approach relies primarily on precursor–product relationships established through controlled laboratory chamber experiments and field campaigns (Zhao et al., 2018; Wang et al., 2021; Deng et al., 2021; Xu et al., 2021; Mutzel et al., 2015; Brüggemann et al., 2020; Yang et al., 2024; Duporté et al., 2020; Huang et al., 2023b; Wang et al., 2022b; Riva et al., 2016a). Based on these established precursor–product relationships, detected OSs and NOSs were classified into four groups: Monoterpene OSs (including Monoterpene NOSs, hereinafter), Aliphatic OSs (including Aliphatic NOSs, hereinafter), Aromatic OSs (including

Aromatic NOSs, hereinafter), and Sesquiterpene OSs (including Sesquiterpene NOSs, hereinafter) (see Table S4 for details). It is apparently that this approach has notable limitations when applied to detected OS in atmospheric aerosols. A substantial fraction of detected OSs does not match known laboratory tracers and are thus labeled Unknown OSs (including Unknown NOSs, hereinafter).

Lines 285–188:

Figures 2(a) and 2(b) shows the average mass concentrations and fractions of different OSs categories across the three cities, based on classification approach based on OSs' elemental composition and laboratory chamber-derived precursor–OS relationships and our precursor-based PMF classification approach developed in this work, respectively (see Section 2.3 for details).

*(24) L237: What is the standard deviation of this number?*

[response]

The standard deviation has been added.

[revised]
Lines 305–306:

High Fe mass concentration was observed in Taiyuan ($0.79 \pm 0.53$ μg/m$^3$), further facilitated the formation of Aromatic OSs.

*(25) L245. It seems unclear why low anthropogenic emissions would promote dominance of aliphatic OSs and NOs. Please clarify or correct.*

[response]

We are sorry for the lack of clarity in our original manuscript. We did not intend to suggest that Aliphatic OSs and NOSs are formed under conditions of absolutely low anthropogenic emissions. The point we wanted to convey is based on a relative comparison among the three cities.

Our data indicates that Beijing experiences a different anthropogenic emission profile compared to Taiyuan and Changsha. Specifically, the emissions of precursors critical for forming monoterpene-derived and aromatic OSs and NOSs are relatively lower in Beijing. This results in a lower absolute and relative contribution of Monoterpene OSs/NOSs and Aromatic OSs/NOSs to the total OS budget in Beijing, as shown in Figure 2 in the main text. Therefore, the relative mass fraction of Aliphatic OSs and NOSs, which are primarily generated from the oxidation of long-chain alkanes (e.g., from vehicle emissions) and SO$_2$, becomes more prominent or "dominant" in Beijing's total OS. This prominence is further amplified by the city's typically lower RH, which inhibits the aqueous phase formation of Monoterpene OSs/NOSs. We have

revised the relevant paragraph in the manuscript to clarify this comparative context in the revised manuscript.

[revised]
Lines 307–317:

The highest total mass fractions of Aliphatic OSs were observed in Beijing (28.1%). Since vehicle emissions, which is an important source of long-chain alkenes (He et al., 2022; Wang et al., 2021; Riva et al., 2016b; Tao et al., 2014; Tang et al., 2020), substantially contributed to $PM_{2.5}$ in all cities (Figure S4), the relative dominance of Aliphatic OSs in Beijing can be attributed to a comparative reduction in the emissions of precursors for Monoterpene OSs and Aromatic OSs. Specifically, Beijing exhibits lower emissions of monoterpene and aromatic VOCs precursors relative to Taiyuan and Changsha, which results in a reduced contribution of Monoterpene and Aromatic OSs to the total OS (see Figure 2(b)). Therefore, the relative mass fraction of Aliphatic OSs, which primarily derived from between sulfate and photooxidation products of alkenes (Riva et al., 2016b), becomes more prominent in Beijing. Additionally, low RH in Beijing further suppresses the aqueous-phase formation of Monoterpene OSs, amplifying the relative importance of Aliphatic OSs.

*(26) L254: Figure 2(a)?*

[response]
Revised.

*(27) L269. Please provide some suggestions or hypotheses.*

[response]
Thanks for your suggestion. Based on our measurement data, we propose a mechanistic hypothesis for this phenomenon. A leading explanation is the dilution of OS's precursors at significantly elevated ALWC: while increased aqueous-phase volume promotes many reactions, it can also dilute the concentrations of OS precursors and intermediates. If OS formation approaches a concentration-dependent plateau, the net production rate may become less sensitive to further ALWC increases, weakening the observed correlation. This hypothesis is consistent with the concurrent evidence of aerosol phase transition (to a fully liquid state at RH > 60%) discussed in the manuscript, where the reactive interface may become saturated.

[revised]
Lines 337–341:

The initial correlation rise reflects ALWC-enhanced sulfate-driven heterogeneous OS formation (Wang et al., 2016a; Cheng et al., 2016), while the decline at elevated RH may due to the

increase in ALWC dilutes the concentrations of precursors and intermediates of Aliphatic OSs within the aqueous phase. Therefore, Aliphatic OSs formation were not further promoted, exhibiting the non-linear response of their mass concentrations and ALWC.

*(28) L288: I did not find data point below pH 1.0.*

[response]

We are sorry for the incorrect statement. We have revised in the manuscript.

*(29) Figure 1:*

*1) Please increase the resolution of the figure for better clarity.*

*2) As the authors argue that increases in ALWC facilitate OS formation through aqueous-phase reactions, could you also discuss why the 31 December sample from Beijing showed the highest organosulfate concentrations despite having low ALWC?*

*3) Please consider replotting the RH data. Currently, a color gradient is used to represent different RH values for each day, but the points are connected with lines that retain the previous point's color. This presentation is somewhat confusing, as it may give the impression that RH remained constant over several days, which is not the case.*

*4) Please state clearly in the figure text that the time scales are different between sites.*

[response]

Thanks for your comments.

1), 3), and 4) Revised.

2) Thanks for your comment. We need to clarify that we did not observed the highest OS mass concentration in Beijing on December 31$^{st}$, but the highest OS/OA ratio. This result does not contradict the facilitating role of ALWC on OS formation, but rather highlights that under specific meteorological and chemical regimes, other formation driving factors can become dominant. Specifically, high Ox concentration was observed on December 31$^{st}$. As discussed in our manuscript (see Section 3.2), enhanced atmospheric oxidation capacity promotes the gas-phase oxidation of VOC precursors and facilitate OS formation. In addition, low aerosol pH was also observed on December 31$^{st}$. High aerosol acidity is a critical driving factor for OS formation (see Section 3.2 and Figure 3(b)). The strong acid-catalysis likely significantly enhanced the heterogeneous formation of OS. We have added relevant discussion in the revised manuscript.

[revised]

Lines 278–284:

It is noteworthy that the single highest OS/OA ratio in Beijing was observed on December 31st under low RH. This phenomena highlights that ALWC, while a major driving factor of OS formation, is not an exclusive control. Specifically, this day showed high atmospheric oxidative capacity and aerosol acidity. We note that under such conditions, efficient acid-catalyzed heterogeneous reactions of gas-phase oxidation products could drive substantial OS formation. The impact of ALWC, atmospheric oxidative capacity, and aerosol pH on OS formation will be discussed in detail in Section 3.2.

Figure 1:

[Figure]

*(30) Figure 2: Please increase the resolution of the figure for better clarity.*

[response]

Revised

[revised]

[Figure]

*(31) Figure 3:*

*1) The color scheme in panels (a) and (b) should be improved, as some colors, particularly the light yellow, are difficult to distinguish.*

*2) Please explain in the figure text that the grey dots are Aliphatic OSs and NOs.*

[response]

Revised

[revised]

[Figure]

**References**

Cheng, Y., Zheng, G., Wei, C., Mu, Q., Zheng, B., Wang, Z., Gao, M., Zhang, Q., He, K., Carmichael, G., Pöschl, U., Su, H.: Reactive nitrogen chemistry in aerosol water as a source of sulfate during haze events in China. Science Advances. 2, e1601530. 10.1126/sciadv.1601530, 2016.

Eddingsaas, N. C., VanderVelde, D. G., Wennberg, P. O.: Kinetics and Products of the Acid-Catalyzed Ring-Opening of Atmospherically Relevant Butyl Epoxy Alcohols. The Journal of Physical Chemistry A. 114, 8106-8113. 10.1021/jp103907c, 2010.

Huang, L., Cochran, R. E., Coddens, E. M., Grassian, V. H.: Formation of Organosulfur Compounds through Transition Metal Ion-Catalyzed Aqueous Phase Reactions. Environmental Science & Technology Letters. 5, 315-321. 10.1021/acs.estlett.8b00225, 2018.

Huang, L., Coddens, E. M., Grassian, V. H.: Formation of Organosulfur Compounds from Aqueous Phase Reactions of S(IV) with Methacrolein and Methyl Vinyl Ketone in the Presence of Transition Metal Ions. ACS Earth and Space Chemistry. 3, 1749-1755. 10.1021/acsearthspacechem.9b00173, 2019.

Iinuma, Y., Müller, C., Berndt, T., Böge, O., Claeys, M., Herrmann, H.: Evidence for the Existence of Organosulfates from β-Pinene Ozonolysis in Ambient Secondary Organic Aerosol. Environmental Science & Technology. 41, 6678-6683. 10.1021/es070938t, 2007.

Riva, M., Da Silva Barbosa, T., Lin, Y. H., Stone, E. A., Gold, A., Surratt, J. D.: Chemical characterization of organosulfates in secondary organic aerosol derived from the photooxidation of alkanes. Atmospheric Chemistry and Physics. 16, 11001-11018. 10.5194/acp-16-11001-2016, 2016.

Wang, G., Zhang, R., Gomez, M. E., Yang, L., Levy Zamora, M., Hu, M., Lin, Y., Peng, J., Guo, S., Meng, J., Li, J., Cheng, C., Hu, T., Ren, Y., Wang, Y., Gao, J., Cao, J., An, Z., Zhou, W., Li, G., Wang, J., Tian, P., Marrero-Ortiz, W., Secrest, J., Du, Z., Zheng, J., Shang, D., Zeng, L., Shao, M., Wang, W., Huang, Y., Wang, Y., Zhu, Y., Li, Y., Hu, J., Pan, B., Cai, L., Cheng, Y., Ji, Y., Zhang, F., Rosenfeld, D., Liss, P. S., Duce, R. A., Kolb, C. E., Molina, M. J.: Persistent sulfate formation from London Fog to Chinese haze. Proceedings of the National Academy of Sciences. 113, 13630-13635. 10.1073/pnas.1616540113, 2016.

Wang, Y., Hu, M., Wang, Y.-C., Li, X., Fang, X., Tang, R., Lu, S., Wu, Y., Guo, S., Wu, Z., Hallquist, M., Yu, J. Z.: Comparative Study of Particulate Organosulfates in Contrasting Atmospheric Environments: Field Evidence for the Significant Influence of Anthropogenic Sulfate and NOx. Environmental Science & Technology Letters. 7, 787-794. 10.1021/acs.estlett.0c00550, 2020.

Wu, Z., Wang, Y., Tan, T., Zhu, Y., Li, M., Shang, D., Wang, H., Lu, K., Guo, S., Zeng, L., Zhang, Y.: Aerosol Liquid Water Driven by Anthropogenic Inorganic Salts: Implying Its Key Role in Haze Formation over the North China Plain. Environmental Science & Technology Letters. 5, 160-166. 10.1021/acs.estlett.8b00021, 2018.

Xu, R., Chen, Y., Ng, S. I. M., Zhang, Z., Gold, A., Turpin, B. J., Ault, A. P., Surratt, J. D., Chan, M. N.: Formation of Inorganic Sulfate and Volatile Nonsulfated Products from Heterogeneous Hydroxyl Radical Oxidation of 2-Methyltetrol Sulfate Aerosols: Mechanisms and Atmospheric Implications. Environmental Science & Technology Letters. 11, 968-974. 10.1021/acs.estlett.4c00451, 2024.

Ye, C., Lu, K., Song, H., Mu, Y., Chen, J., Zhang, Y.: A critical review of sulfate aerosol formation mechanisms during winter polluted periods. Journal of Environmental Sciences. 123, 387-399. https://doi.org/10.1016/j.jes.2022.07.011, 2023.

Zheng, B., Zhang, Q., Zhang, Y., He, K. B., Wang, K., Zheng, G. J., Duan, F. K., Ma, Y. L., Kimoto, T.: Heterogeneous chemistry: a mechanism missing in current models to explain secondary inorganic aerosol formation during the January 2013 haze episode in North China. Atmos. Chem. Phys. 15, 2031-2049.

10.5194/acp-15-2031-2015, 2015.

---

## Author Response (AR2)

**Response to Referee#3**

Thanks to your careful reading and their constructive comments and suggestions on our manuscript. The reviewers' comments and suggestions are shown as *italicized font*, our response to the comments is normal font. New or modified text is in normal font and in blue. Details are as follows.

▪▪▪▪▪▪▪▪▪▪▪▪▪▪▪▪▪▪▪▪▪▪▪▪▪▪▪▪▪▪▪▪▪▪▪▪▪▪▪▪▪▪▪▪▪▪▪▪▪▪

Referee's comments:

*The revised manuscript has addressed the majority of my concerns. Upon addressing the remaining few points below, I recommend it for publication in ACP.*

*In general, the new text needs to be checked for correct language, preferably by a native English-speaking person.*

[response]

Thanks for your comments. Please check our point-by-point response.

*General comments:*

*(1) The conclusions regarding wintertime OS underestimation appear reasonable, but the wording could be softened more to avoid implying representativeness for the entire winter season based on a short campaign.*

[response]

Thank you for your suggestion. We fully agree that the representativeness of a several-week filed campaign for the entire winter season should be carefully qualified. Our intention was to highlight a significant pattern observed during our study period, not to claim it is universally representative of all winter conditions.

In the revised manuscript, we have carefully the description throughout the manuscript, particularly in Section 4. The key findings in this work, such as the underestimation of Aliphatic OSs and the role of specific formation driving factors as characteristics observed in our winter campaign. This clarifies that the conclusions are specific to the conditions we measured, while maintaining the strength and importance reported in this work.

[revised]

Line 36-37:

The formation driving factors of Aliphatic OSs during the field campaign were further investigated.

Line 41-44:

These results reveal a significant underestimation of OSs derived from anthropogenic emissions in wintertime, particularly Aliphatic OSs, highlighting the need for a deeper

Line 417-420:

In addition, our field campaigns were conducted in three typical different Chinese cites, the effect of these driving factors on the formation of Aliphatic OSs may not be applicable to other cities with different atmospheric conditions.

*(5) Thank you for providing more details about the sampling. For how long were the filters prebaked? What was the brand of the RH-resolved sampler?*

[response]

Thanks for your comment. The filters were prebaked for 9 hours under the temperature of 550 °C. The RH-resolved sampler is a home-made instrument, the picture of this RH-sampler is shown in Figure R1. The ambient RH is monitored using an RH sensor connected to this instrument. Before sampling, the switch point is set on the control computer, and the RH sensor measures the ambient RH with the time resolution of 1 second during sampling. The sampling channel is selected based on the range of ambient RH. The real-time cumulative sampling volume of each channel is recorded. When sampling terminates, the cumulative sampled volume for each channel, and real-time ambient RH can be exported for further data analysis.

[Figure]

**Figure R1** The picture of home-made RH-resolved sampler used in this work

[revised]
Line 108-109:

All quartz fiber filters were pre-baked at 550 °C for 9 hours before sampling to remove the background organic matters.

*Specific Comments:*

*(11) The explanation of site selection based on meteorology and emission characteristics is clear and well justified. But I think the discussion could be further strengthened by briefly clarifying the extent to which the inferred OS formation mechanisms are expected to be generalizable beyond these three cities, and under what conditions they may not apply. This would help readers better understand the broader applicability and limitations of the conclusions.*

[response]

Thanks for your suggestion. We agree that clarifying the scope of applicability is crucial. The key driving factors identified in this work, including ALWC, inorganic sulfate, and $O_x$, are likely operative in other urban environments sharing similar winter conditions characterized by high anthropogenic emissions and moderate-to-high humidity, however, their relative importance and manifestation may differ under contrasting scenarios. In the revised manuscript, we have discussed conditions where our conclusions might not directly apply, such as in summer with strong biogenic emissions, in regions with low aerosol acidity, or in arid cities with persistently low RH.

[revised]
Line 394-399:

It should be noted that though formation driving factors of Aliphatic OSs identified in this work, including ALWC, inorganic sulfate, and $O_x$, are likely applicable in other urban environments sharing similar winter conditions characterized by high anthropogenic emissions and moderate-to-high humidity. However, their importance may differ in other cities with different atmospheric conditions, like in summer with strong biogenic emissions, in regions with low aerosol acidity, or in arid cities with persistently low RH.

*(14) Please note the typo in redisslove -> re-dissolve or re-dissolving*

[response]

Revised.

*The authors have adequately addressed the remaining comments.*

*All figures have also been appropriately revised.*

*In line 23 of the new manuscript "precursor-OS correspondence". It is unclear what you mean, but I assume you just mean "precursor" of the OS.*

[response]

We have revised relevant description.

[revised]

Line 25-27:

However, molecular composition, the relationship between OSs and their precursors, and formation driving factors of OSs at different atmospheric conditions have not been fully constrained.